



# Year-round record of near-surface ozone and "O₃ enhancement events" (OEEs) at Dome A, East Antarctica

Minghu Ding[1,2,*], Biao Tian[1,*], Michael C. B. Ashley[3], Davide Putero[4], Zhenxi Zhu[5], Lifan Wang[5], Shihai Yang[6], Chuanjin Li[2], Cunde Xiao[2,7]

[1]State Key Laboratory of Severe Weather, Chinese Academy of Meteorological Sciences, Beijing 100081, China
[2]State Key Laboratory of Cryospheric Science, Northwest Institute of Eco-Environment and Resources, Chinese Academy of Sciences, Lanzhou 730000, China
[3]School of Physics, University of New South Wales, Sydney 2052, Australia
[4]CNR–ISAC, National Research Council of Italy, Institute of Atmospheric Sciences and Climate, via Gobetti 101, 40129, 10   Bologna, Italy
[5]Purple Mountain Observatory, Chinese Academy of Sciences, Nanjing 210034, China
[6]Nanjing Institute of Astronomical Optics & Technology, Chinese Academy of Sciences, Nanjing 210042, China
[7]State Key Laboratory of Earth Surface Processes and Resource Ecology, Beijing Normal University, Beijing 100875, China
*These authors contributed equally to this work.
*Correspondence to*: Minghu Ding (dingminghu@foxmail.com)

**Abstract.** Dome A, the summit of the east Antarctic Ice Sheet, is an area challenging to access and is one of the harshest environments on Earth. Up until recently, long term automated observations from Dome A were only possible with very low power instruments such as a basic meteorological station. To evaluate the characteristics of near-surface $O_3$, continuous observations were carried out in 2016. Together with observations at the Amundsen-Scott Station (South Pole – SP) and 20   Zhongshan Station (ZS, on the southeast coast of Prydz Bay), the seasonal and diurnal $O_3$ variabilities were investigated. The results showed different patterns between coastal and inland Antarctic areas that were characterized by high concentrations in cold seasons and at night. The annual mean values at the three stations (DA, SP and ZS) were $29.2 \pm 7.5$ ppb, $29.9 \pm 5.0$ ppb and $24.1 \pm 5.8$ ppb, respectively. We investigated the effect of specific atmospheric processes on near-surface summer $O_3$ variability, when $O_3$ enhancement events (OEEs) are systematically observed at DA (average monthly frequency peaking 25   up to 64.5% in December). As deduced by a statistical selection methodology, these $O_3$ enhancement events (OEEs) are affected by a significant interannual variability, both in their average $O_3$ values and in their frequency. To explain part of this variability, we analyzed the OEEs as a function of specific atmospheric processes: (i) the role of synoptic-scale air mass transport over the Antarctic Plateau was explored using the Lagrangian back-trajectory analysis – Hybrid Single-Particle Lagrangian Integrated Trajectory (HYSPLIT) method and (ii) the occurrence of "deep" stratospheric intrusion events was 30   investigated using the Lagrangian tool STEFLUX. The specific atmospheric processes, including synoptic-scale air mass transport, were analysed by the HYSPLIT back-trajectory analysis and the potential source contribution function (PSCF) model. Short-range transport accounted for the $O_3$ enhancement events (OEEs) during summer at DA, rather than efficient local production, which is consistent with previous studies of inland Antarctica. Moreover, the identification of recent (i.e.,





4-day old) stratospheric intrusions events by STEFLUX suggested that "deep" events only had a minor influence (up to 1.1

% of the period, in August) on "deep" events during the variability of near-surface summer $O_3$ at DA. The "deep" events during the polar night were significantly higher than those during the polar day. This work provides unique information on ozone variation at DA and expands our knowledge of such events in Antarctica. Data are available at https://doi.org/10.5281/zenodo.3923517 (Ding et al., 2020).

**Key words**: near-surface $O_3$; Antarctica; OEE; STT;

## 1 Introduction

Ozone ($O_3$) is a natural atmospheric component that is found both in the stratosphere and troposphere and plays a major role in the atmospheric environment through radiative and chemical processes. $O_3$ does not have direct natural sources such as emission from the ground or vegetation, but rather is produced in the atmosphere, and its concentration ranges from a few ppb near the Earth's surface to approximately a few ppm in the stratosphere. Stratospheric $O_3$, which is produced as a result

of the photolysis of molecular oxygen, forms a protective layer against the UV radiation from the Sun. By contrast, throughout the troposphere and at the surface, $O_3$ is considered a secondary short-lived air pollutant (Monks et al., 2015), and $O_3$ itself is a greenhouse gas, such that a reduction in concentration has a direct influence on radiative forcing (Mickley et al., 1999; IPCC, 2013; Stevenson et al, 1998).

$O_3$ photochemical production in the troposphere occurs via hydroxyl radical oxidation of carbon monoxide (CO), methane

($CH_4$) and non-methane hydrocarbons (generally referred to as NMHC) in the presence of nitrogen oxides (NOx) (Monks et al., 2015). As these precursors are localized and their lifetimes are generally short, the distribution of near-surface$O_3$, which is produced from anthropogenic precursors, is also localized and time-variant. In the presence of strong solar radiation with $\lambda$ < 424 nm, volatile organic compounds (VOCs) and NOx (NO + $NO_2$), $O_3$ is photochemically produced and can accumulate to a hazardous level during favourable meteorological conditions (Davidson, 1993; Wakamatsu et al., 1996). In the case of

NOx-rich air, $NO_2$ is produced and accumulates via the reaction between NO and $HO_2$ or $RO_2$ (peroxy radicals), which is followed by the accumulation of $O_3$. However, in the case of NOx-poor air, these proxy radicals react with $O_3$ and lead to $O_3$ loss (Lin et al., 1988). Experiments conducted in Michigan (Honrath et al., 2000a) and Antarctica (Jones et al., 2000) found that NOx can be produced in surface snow. This production appears to be directly driven by incident radiation and photolysis of nitrate deposited in the snow (Honrath et al., 2000a, b).

Previous studies have shown that the near-surface $O_3$ of Antarctica may be influenced by a number of climate-related variables (Berman et al., 1999), such as the variation of UV flux caused by the variation of $O_3$ column concentration over Antarctica (Jones and Wolff, 2003; Frey et al., 2015), the accumulation and transport of long-distance, high concentration air masses (e.g., Legrand et al., 2016), and the depth of continental mixing layers. Many studies has observed summer episodes of "$O_3$ enhancement events" (OEEs) in the Antarctic interior (e.g., Crawford et al., 2001; Legrand et al., 2009; Cristofanelli





et al., 2018), and they attributed this phenomenon to the NOx emissions from snowpack and subsequent photochemical $O_3$ production (for example, Jones et al., 2000). Moreover, this may provide an input source for the entire Antarctic region (for example, Legrand et al., 2016; Bauguitte et al., 2011). Indeed, Helmig et al. (2008a,b) provided further insight into the vigorous photochemistry and $O_3$ production that result from the highly elevated levels of NOx in the Antarctic surface layer. During stable atmospheric conditions, which are typically observed during low wind and fair sky conditions, $O_3$ accumulated

in the surface layer can reach up to twice its background concentration. Neff et al. (2008a) showed that shallow mixing layers associated with light winds and strong surface stability can be among the dominant factors leading to high NO levels. As shown by Cristofanelli et al. (2008) and Legrand et al. (2016), the photochemically-produced $O_3$ in the PBL over the Antarctic Plateau can affect the $O_3$ variability thousands of km away from the emission area, due to air mass transport.

The near-surface $O_3$ concentrations at high-elevation sites can also be increased by the downward transport of $O_3$-rich air

from the stratosphere during deep convection and stratosphere-to-troposphere transport (STT) events. Moreover, the stratospheric $O_3$ in the polar regions can be transferred to the troposphere not only during intrusion events but also as a result of slow but prolonged subsidence (e.g., Gruzdev et al., 1993; Roscoe et al., 2004; Greenslade et al. 2017). The earliest studies, carried out by the aircraft flight NSFC-130 over the Ellsworth Mountains of Antarctica in 1978, found that mountainous terrain may induce atmospheric waves that propagate through the tropopause. The tropospheric and

stratospheric air may be mixed, leading to an increase in the tropospheric $O_3$ concentration (e.g., Robinson et al., 1983). Radio soundings at the Resolute and Amundsen-Scott Stations also showed the existence of transport from the stratosphere to the troposphere, and the flux could reach up to $5 \times 10^{10}$ mol/cm$^2$/s (e.g., Gruzdev et al., 1993). Recently, Traversi et al. (2014, 2017) suggested that the variability of air mass transport from the stratosphere to the Antarctic Plateau could affect the nitrate content in the lower troposphere and the snowpack.

Currently, the climatology of tropospheric $O_3$ over Antarctica is relatively understudied because observations of year-round near-surface $O_3$ have been tied to manned research stations. These stations are generally located in coastal Antarctica, except for the South Pole (SP) and Dome C continental stations on the East Antarctic Plateau. Thus, the only information currently available for the vast region between the coast and plateau are spot measurements of boundary layer $O_3$ during summer from scientific traverses (e.g., Frey et al., 2015) or airborne campaigns (e.g., Slusher et al., 2010). Moreover, the vertical profile of

$O_3$ in the troposphere cannot be measured by satellites because the high density of $O_3$ in the stratosphere leads to the inaccurate estimation of tropospheric $O_3$ by limb-viewing sensors. Estimates of total $O_3$ in the troposphere have been made by subtracting the stratospheric $O_3$ column (determined by a limb-viewing sensor) from the total column of $O_3$ (measured by a nadir-viewing sensor) (Fishman et al., 1992). In other words, tropospheric profiles cannot be obtained by satellites, and we cannot examine the spatial distribution of near-surface $O_3$ from space. As a result of these limitations, a dearth of information

exists regarding the spatial gradient of near-surface $O_3$ across Antarctica and how it varies throughout the year.

To better understand the spatial variations and the source-sink mechanisms of near-surface $O_3$ in Antarctica, near-surface $O_3$ concentrations were measured during 2016 at Dome A (DA) and the Zhongshan Station (ZS). Together with records from





the Amundsen-Scott Station (SP), we analysed specific processes that affect the intra-annual variability in surface $O_3$ over the East Antarctic Plateau; in particular, we determined (i) the synoptic-scale air mass transport within the Antarctic interior and (ii) the role of STT transport. This study broadens the understanding of the spatial and temporal variations in the near-surface $O_3$ concentration and transport processes that impact tropospheric $O_3$ over high plateaus.

## 2 Sites and methods description

### 2.1 Near-surface ozone observations

The Kunlun Station (80°25'02"S, 77°06'59"E, altitude 4087 m) is located in the DA area, on the summit of the east Antarctic Ice Sheet (Figure 1). The $O_3$ monitor is located at the PLATO Antarctic site testing observatory. The instrument was powered by the PLATO-A observatory, an improved version of the PLATO observatory described by Lawrence et al. (2009), which also provided internet access via the Iridium satellite network. Due to the limitation of energy consumption and conditions encountered during transportation, the Thermo 49i $O_3$ monitor cannot be used. A portable $O_3$ monitor with low energy consumption is suitable. On the 1st of Jan 2016, we deployed a Model 205 Dual Beam Ozone Monitor during the 33rd Chinese National Antarctic Research Expedition. The Model 205 Dual Beam Ozone Monitor makes use of two detection cells to improve precision, baseline stability, and response time. In the dual beam instrument, UV light intensity measurements $I_0$ ($O_3$-scrubbed air) and $I$ (unscrubbed air) are performed simultaneously. Combined with other improvements, this instrument is the fastest UV-based $O_3$ monitor on the market, with such small size, weight, and power requirements characteristics (Table 1). Fast measurements are particularly desirable for unattended stations, aircraft and balloon measurements, where high spatial resolution is desired. The Model 205 Dual Beam Ozone Monitor (205 2B) is an Environmental Protection Agency (EPA) federal equivalent method (FEM).

The Zhongshan Station (69°22'12"S, 76°21'49"E, altitude 18.5 m) is located at the edge of the east Antarctic Ice Sheet (Figure 1), where we installed a UV absorption near-surface $O_3$ analyzer (EC9810A) for long-term near-surface $O_3$ monitoring. The observational frequency was 3 min, and the data were transferred in real time to Beijing. Furthermore, to prevent data losses, a CR1000 data logger was used to record the data output in real time. Every three months, the $O_3$ analyzer was calibrated using the EC9811 $O_3$ calibrator, and 5 standard concentrations of $O_3$ gas were generated for each calibration. The calibration concentration and measured concentration underwent correlation analysis, and seasonal calibration results were generated every three months. In 2016, 5 calibrations were made, and the appropriate correlation coefficients (r) were all greater than 0.9995.

The Amundsen-Scott Station (89°59'51.19 "S, 139°16'22.41" E, altitude 2835 m) is located at the SP and operated by the United States. The near-surface $O_3$ data were downloaded from the Earth System Research Laboratory Global Monitoring Division under the NOAA (https://www.esrl.noaa.gov/gmd/dv/data).



The hourly data of these stations collated here are available at https://doi.org/10.5281/zenodo.3923517 (Ding et al., 2020).

## 2.2 Calibration process and results

Generally, the zero point, span point and operation parameters of the $O_3$ monitor should be checked before each operation. The zero point should be checked regularly during continuous observation. While such regular calibration was done at the Global Atmosphere Watch (GAW) and Zhongshan Station, it was not possible at DA due to the lack of logistic support and the extreme environment. To minimize the error and evaluate the accuracy of the experiment, a UV-absorption $O_3$ calibrator Thermo 49i-PS was used to examine the Model 205. The calibration procedure follows China's environmental protection
standard "ambient air - Determination of ozone - ultraviolet method" (HJ590-2010) (http://www.mee.gov.cn/gkml/sthjbgw/sthjbgg/201808/t20180815_451411.htm) which is more strict than USEPA (Ref: USEPA .     Quality Assurance Handbook for Air Pollution Measurement Systems Volume II: Ambient Air Quality Monitoring Program[EB/OL].[2008-12-01]. http://www.epa.gov/ttn/amtic/files/ambient/pm25/qa/ QAHandbook-Vol-II.pdf.): the slope of calibration curve ranges between 0.95-1.05, and the intercept ranges between -5-5 ppb. Instruments used in the
calibration process include a DOA-p512-bn air compressor (USA), in addition to the Thermo 49ips $O_3$ calibrator and the Model 205 $O_3$ monitor. Before each test, the $O_3$ calibrator and the $O_3$ monitor were turned on and preheated for 12 hours, and the measuring range was set to 400 ppb. We first generate a zero concentration using the Thermo 49ips and, once the analyzer response has stabilized on zero reading, we adjusted the Model 205's internal zero setting to matches the zero air source. Then, $O_3$ airflow at 400 ppb level was generated and injected into the analyzer, and a correction factor was calculated
based on the observed value, which was then loaded into the Model 205 configuration.

After the calibration of the internal zero/span settings, a second stage of calibration was performed involving multi-point verification to check the response and stability of the analyzer. On Oct 5th 2015 (before the instrument was shipped) and May 6th 2017(the day that the instrument was transported back from Antarctica), a zero and 7 upscale points (0, 20, 35, 50, 65, 80, 100, 120 ppb) encompassing the full scale of the observation range (Table 2), were generated by the Thermo 49ips to test the
Model 205 analyzer. Each point was observed for 15 min, during the last 10 minutes of which readings were taken every minute of the calibrator and analyzer. Based on this experiment, the slope and intercept of the calibration curve were calculated by least squares. The results are shown in Table 2, it can be concluded that the slopes of the linear correction curve were 0.99936 and 1.02520, and the intercepts were 0.53861 and 0.85220l (Table 3), which fulfilled the requirements of HJ590-2010 and USEPA.

Another challenge when monitoring the atmosphere is the stability of the analyzer, which includes the analyzer's response time. Similarly with the regular calibration, it could not be performed during the observation period, but it was reassuring that the Model 205 was still in good condition when we did the multi-point verification in May 2017, as shown in Table 3.



The slope and intercept of the two calibration curves changed little and the standard uncertainties were small. To further test the stability, data consistency was also examined and the mean absolute deviation between two adjacent values was only

0.09 ppb. The largest difference was 0.61 ppb, indicating that the analyzer was stable and reliable.

Before analysis, a variance test was used to remove abnormal data based on the Laida criterion method, which assumes that the records obeyed a normal distribution. The formula is $\left| x_i - \overline{x} \right| > 3\sigma$, where $x_i$ is the measured value, $\overline{x}$ is the time series mean and $\sigma$ is the standard deviation. After processing, 99.2%, 91.6%, and 99.5% of the hourly mean data were retained from the Amundsen-Scott Station, Zhongshan Station and Kunlun Station, respectively.

**2.3 Air mass back-trajectory calculations**

Gridded meteorological data for backward trajectories in Hybrid Single-Particle Lagrangian Integrated Trajectory (HYSPLIT) were obtained from the Global Data Assimilation System (GDAS-1) operated by the U.S. National Oceanic and Atmospheric Administration (NOAA) with 1°×1° horizontal resolution and 23 vertical levels, from 1000 hPa to 20 hPa (http://www.arl.noaa.gov/gdas1.php).

The HYSPLIT backward air mass trajectory model was previously applied to atmospheric research in Antarctica (Legrand et al., 2009; Hara et al., 2011). We used the HYSPLIT model in this paper to analyse the impact of varying air mass sources and the intrusion of stratospheric $O_3$. Backward trajectories and clusters were calculated using the US National Oceanic and Atmospheric Administration (NOAA)-HYSPLIT model (Draxler and Rolph, 2003; http://ready.arl.noaa.gov/HYSPLIT.php), which is a free software plug-in for MeteoInfo (Wang, 2014; http://meteothink.org/). The backward trajectories starting

height was set at 20 m above the surface and the total run times was 120 hours for each backward trajectory, and each run was performed in time intervals of 6 hours (00:00, 06:00, 12:00, 18:00).

The integral error part of the trajectory calculation error can be estimated by simulating the backward trajectory at the end of the forward trajectory and comparing the differences of the tracks. The starting point of the backward integration is set as (77.12 ° E, 80.42 °S, 20m a.g.l.), the backward integration is 120 hours. Then the point reached at this time is taken as the

starting point, and a forward simulation is made for 120h. In this simulation experiment, the contribution of integration error to trajectory calculation error is very small within the first 72 hours. With the extension of integration time, the integration error slightly increases.

**2.4 Potential source contribution function**

The observation of a secondary maximum of $O_3$ in November–December at the inland Antarctic sites was first reported for

the SP by Crawford et al. (2001), and was attributed to photochemical production induced by high NOx levels in the atmospheric surface layer, which were generated by the photo-denitrification of the Antarctic snowpack (same as Davis et al.,



2001). At DC, a secondary maximum in November–December 2007 was also reported by Legrand et al. (2009), proving that photochemical production of $O_3$ in the summer takes place over a large part of the Antarctic Plateau. A further study by Legrand et al. (2016) found that the highest near-surface $O_3$ summer values were observed within air masses that spent
extensive time over the highest part of the Antarctic Plateau before arriving at DC. To investigate the possible influence of synoptic-scale air mass circulation on the occurrence of OEEs at DA, 5-day HYSPLIT back-trajectories were analyzed (Figure 9). We used the potential source contribution function (PSCF, see, e.g., Hopke et al., 1995; Brattich et al., 2017) to calculate the conditional probabilities and identify the geographical regions related to the occurrence of NOEEs and OEEs at DA (Figure 7).

As in Yin et al. (2017), the potential source contribution function (PSCF) assumes that back trajectories arriving at times of high mixing ratios likely point to significant pollution directions (Ashbaugh et al., 1985). This function was often applied to locate air masses associated with high levels of near-surface $O_3$ at different sites (Kaiser et al., 2007; Dimitriou and Kassomenos, 2015). In this study, the PSCF was calculated using HYSPLIT trajectories. The top of the model was set to 10000 m a.s.l. The PSCF values for the grid cells in the study domain were calculated by counting the trajectory segment
endpoints that terminated within each cell (Ashbaugh et al., 1985). If the total number of end points that fall in a cell is $n_{ij}$ and there are $m_{ij}$ points for which the measured $O_3$ parameter exceeds a criterion value selected for this parameter, then the conditional probability, the PSCF, can be determined as

$$\text{PSCF}_{ij} = \frac{m_{ij}}{n_{ij}} \qquad (1)$$

The concentrations of a given analyte greater than the criterion level are related to the passage of air parcels through the ijth
cell during transport to the receptor site. That is, cells with high PSCF values are associated with the arrival of air parcels at the receptor site, which has near-surface $O_3$ concentrations that are higher than the criterion value. These cells are indicative of areas with 'high potential' contributions of the constituent. Identical PSCFij values can be obtained from cells with very different counts of back-trajectory points (e.g., grid cell A with $m_{ij}$ = 5000 and $n_{ij}$ = 10000 and grid cell B with $m_{ij}$ = 5 and $n_{ij}$ = 10). In this extreme situation, grid cell A has 1000 times more air parcels passing through it than grid cell B.
Because the particle count in grid cell B is sparse, the PSCF values in this cell are highly uncertain. To explain expound the uncertainty due to the low values of $n_{ij}$, the PSCF values were scaled by a weighting function $W_{ij}$ (Polissar et al., 1999). The weighting function reduced the PSCF values when the total number of endpoints in a cell was less than approximately 3 times the average number of end points per cell. In this case, Wij was set as follows:

$$W_{ij(NOEE)} = \begin{cases} 1.00 & nij > 12\,Nave \\ 0.70 & 12\,Nave > nij > 3\,Nave \\ 0.42 & 3\,Nave > nij > 1.5\,Nave \\ 0.05 & Nave > nij \end{cases} \qquad (2)$$



$$W_{ij(OEE)} = \begin{cases} 1.00 & nij > 8\,Nave \\ 0.70 & 8\,Nave > nij > 2\,Nave \\ 0.42 & 2\,Nave > nij > 1\,Nave \\ 0.05 & Nave > nij \end{cases} \qquad (3)$$

where Nave represents the mean nij of all grid cells. The weighted PSCF values were obtained by multiplying the original
PSCF values by the weighting factor.

## 3 Near-surface $O_3$ variability

### 3.1 Mean concentration

At the DA, SP, and ZS sites, the annual mean molar ratios of near-surface $O_3$ were $29.2 \pm 7.5$ ppb, $29.9 \pm 5.0$ ppb and $24.1 \pm 5.8$ ppb, respectively; the maximum annual mean molar ratio reached 42.5 ppb, 46.4 ppb and 32.8 ppb, respectively; and the minimum annual mean molar ratios were 14.0 ppb, 10.9 ppb and 9.9 ppb, respectively. The inland stations are characterized by higher annual mean molar ratios than the coastal station.

There were also obvious differences between polar day and polar night at all stations. In Figure 2, we define the polar day
and night windows by the day of year margins and have used different shading colours to identify the polar day and polar night. The average molar ratios of near-surface $O_3$ during polar night at the DA, SP and ZS sites were $34.1 \pm 4.3$ ppb, $31.5 \pm 3.9$ ppb and $28.7 \pm 1.3$ ppb, respectively, and much lower concentrations appeared during non-polar night, with corresponding values of $26.1 \pm 7.0$ ppb, $28.1 \pm 5.8$ ppb and $23.1 \pm 5.9$ ppb, respectively. Interestingly, the SP had the highest near-surface $O_3$ concentration during non-polar night, whereas at DA the highest concentration occurred during polar night
and the largest variation occurred at this site.

### 3.2 Seasonal variation

In this part, we define Oct-Mar as the warm season and Apr-Sept as the cold season, which is similar to the definition of polar day and night.

In agreement with previous studies (Oltmans et al., 1976; Gruzdev et al., 1993; Ghude et al., 2005), the concentrations of
near-surface $O_3$ at the three stations were high and less variable during the cold season and low and more variable during the warm season (Figure 3). In Antarctica, the emissions of $O_3$ precursors are generally less than those at mid and low latitudes, whereas ultraviolet radiation is relatively strong; thus, when solar radiation occurs, the depletion effect is much greater than the effects from photochemical reactions during the warm season (Schnell et al., 1991). As explained by previous studies, during the polar night, due to the lack of light, the photochemical reactions stop. Moreover, due to the lack of loss effect, the



$O_3$ concentration gradually increased and the fluctuations became smaller. During the polar night, the monthly variation of surface $O_3$ at ZS was lower than that at the DA but higher than that at the SP. However, due to strong UV radiation in the low latitude areas and the presence of bromine-controlled $O_3$ depletion events in coastal areas, the ZS shows a large seasonal variations during the non-polar night (Wang et al., 2011; Prados-Roman et al., 2017). However, at the SP Station, the largest standard deviation was observed in December, similarly to the characteristics at Dome-C station (DC) from November to

December (Legrand et al., 2009; Cristofanelli et al., 2018). Figures 2 and 3 indicate that the near-surface $O_3$ showed obviously larger variations at the DA than the SP during the polar night, since, due to the different geographical location, the meteorological conditions of DA and SP are different. The abnormal fluctuation of $O_3$ concentration over the DA during the polar night may be related to its special geographical environment.

   As mentioned in the introduction section, mountainous topography/mountain waves may disturb advection transport in the

stratosphere and lead to downward transportation to the troposphere (Robinson et al., 1983). DA is on the summit of the east Antarctic Ice Sheet, and the tropospheric depth is only ~4.6 km (Liang et al., 2015), which favours exchange between the stratosphere and troposphere. However, the topography in this area is very flat and creates a disadvantage for mountain waves. Does $O_3$ transport occur? We will analyse and discuss this question in section 4.

### 3.3 Diurnal variation

To characterize the typical monthly $O_3$ diurnal variations at the three stations, we analysed the mean diurnal variations of $O_3$ at the three stations (Figure 4) and the standard deviation of the mean diurnal variations (Figure 5). At the DA site, the mean diurnal concentrations for each month were relatively steady, with the standard deviation of the mean diurnal concentration for each month being lower than 0.4 ppb. At the SP, the mean diurnal concentrations were less variable as well. Except for December, the standard deviation of the mean diurnal concentration was lower than 0.3 ppb. At ZS, except for October, the

standard deviation of the mean diurnal concentration was greater than that at the other two stations. In particular, the standard deviation of the mean diurnal concentration of ZS in September, November and December exceeded 0.5 ppb. The mean diurnal variations in different time periods were not obvious, and the mean diurnal concentrations of the three stations fluctuated within a range of less than 1 ppb, indicating that daily photochemistry reactions were not the dominant factor in near-surface $O_3$ at the three stations. The magnitude of the diurnal variation was low, which is similar to the variations found

at other Antarctic stations (Gruzdev et al., 1993; Ghude et al., 2005; Oltmans et al., 2008).



## 4 Ozone under OEEs at the Kunlun Station

### 4.1 Identification of OEEs

Our method to select the days characterized by OEEs is based on the procedure used in Cristofanelli et al. (2018). First, a sinusoidal fit is used to calculate the $O_3$ annual cycle not affected by the OEEs, then a probability density function (PDF) of
the deviations from the sinusoidal fit is calculated, with the application of a Gaussian fit to the obtained PDF. As reported in Giostra et al. (2011), the deviations from the Gaussian distribution (calculated by using the Origin 9© statistical tool) can be used to identify observations affected by non-background variability. We computed the further Gaussian fitting of PDF points beyond 1 σ (standard deviation) of the Gaussian PDF, and determined the non-background $O_3$ daily values that may be affected by "anomalous" $O_3$ enhancement. The intersection of the two fitting curves is taken as our screening threshold (3.4
ppb at SP, 3.4 ppb at Da and 2.5 ppbzs at ZS). Figures 6a, 6b and 6c show OEE days and NOEE days at these three stations, while Figures 6d, 6e and 6f report the distribution frequency of OEE days.

In total, 42 days at DA were found to be affected by anomalous OEEs: 14.3% in January, 2.4% in May, 14.3% in June, 4.8% in July, 11.9% in August, 4.8% in November and 47.6% in December (Figure 6e, blue bars). This result clearly indicates that half of the anomalous days occurred in December, followed by January and June. At SP, 36 days with OEEs were found in
2016: 44.4% in January, 30.6% in November, and 25% in December (Figure 6d, grey bars). Apparently, OEEs occur only in summertime at this measurement site. ZS was characterized by more days with OEEs: 53 days in April (34.0%), followed by September (18.9%), January (13.2%), October (11.3%), November (11.3%), December (5.7%) March (3.8%) and May (1.9%) (Figure 6f, yellow bars).

From the results above, SP was characterized by concentrated OEE occurrences, and ZS had the most scattered OEEs pattern.
In addition, all OEEs at SP and ZS occurred during the Antarctic warm season, and no OEEs were present during the polar night, similarly to the pattern observed at DC (Cristofanelli et al., 2018). In contrast, the OEEs also occurred during the polar night in DA, and the number of OEE occurrence days accounted for up to 33% of the total number of events throughout the year. Previous studies (e.g., Legrand et al., 2016; Cristofanelli et al., 2018) carried out in DC showed that the $O_3$ variability at DC could be associated with processes occurring at long temporal scales. In addition, the accumulation of
photochemically produced $O_3$ during transport of air masses was the main reason for OEEs, whereas the stratospheric intrusion events had only a minor influence on OEEs (up to 3%). This finding cannot explain the temporal occurrence pattern of OEEs at DA. To determine the unknown cause, we investigated the synoptic-scale air mass transport and the STT occurrence at the measurement site.





## 4.2 Role of synoptic-scale air mass transport

During NOEEs, the air masses arriving at DA mainly come from the west and east of DA, and the 3-D clusters show that the air masses travelled over the Antarctic plateau before reaching DA (Fig. 8b). The difference in the number of the three cluster trajectories is small, and the difference in the corresponding cluster average concentrations is not large. Using the PSCF results, we have identified air masses associated with higher surface ozone at DA during NOEEs (Fig. 8a). The Antarctic Plateau to the east and west of DA had high PSCF weight values (Figure 7), which shows that, during NOEEs, the

potential source area of surface $O_3$ for DA is mainly in the inland plateaus in the east and west, and the area of high PSCF weight values distribution in the east is more larger than other directions.

Compared with NOEEs, the clustering results of trajectories during OEEs have different characteristics. In OEEs, the air masses that arrived at DA were predominantly from the north and from the west, and the 3-D clusters indicated that the 73% of the air mass trajectories came from the area north of DA (red line in Fig. 9a). The average concentrations of the three

clusters differ greatly (Fig. 9c), but they are all higher than those obtained for NOEEs. It should be noted that 68% of Line-2 cluster (green line in Fig. 9a) occurred during the polar night (Fig. 10) and had a high average $O_3$ concentration (reached 36.3 ppb). This shows that the OEEs of the polar night are more affected by the high value $O_3$ air masses over the plateau west of DA than those during the polar day. Using the PSCF results, during OEEs, we did not find a large area of high WPSCF value, the high WPSCF value only appeared in the east and the north of DA over a limited area. However,

independently on the polar day or on the polar night, the Line-1 cluster trajectory accounted for more than 60% during OEEs. In addition, the short distance of Line-1 cluster trajectory indicates that the air mass transport speed is slow, which is conducive to the accumulation of $O_3$ along the way. It can be seen from Figure 9b that the characteristic values of backward trajectory clustering during OEEs are mostly lower than 200 m a.g.l. (supporting the role of snow as the source of near surface $O_3$). As Fiebig et al. (2014) have proposed, the increase of $O_3$ values in the near surface of central Antarctica may

also be related to the transport of free tropospheric air and aged pollution plumes from low latitudes. In addition, Figure 11 shows that the average $O_3$ growth rate reached 0.29 ppb/h during OEEs, while the average $O_3$ growth rate was -0.06 ppb/h during NOEEs (Figure 11). The statistical scatter distribution showed that 97% of OEEs occurred when the wind speed was lower than 4 m/s. The overall average wind speed during OEEs is also significantly lower than that of NOEEs. As Helmig et al. (2008a) have proposed, during stable atmospheric conditions (which typically existed during low wind and fair sky

conditions) ozone accumulates in the surface layer and its concentration increases rapidly.

This finding confirms that the OEEs of DA are mainly caused by the accumulation of high concentrations of air masses transported occurring nearby, and the synoptic-scale transport can favor the photochemical production and the accumulation of $O_3$ accumulation by air masses travelling over the plateau near the north of DA before their arrival.



### 4.3 Role of STT events

4.3.1 Identification of STT events

Several methods can be applied to study stratosphere-to-troposphere transport (STT) events. One method is the chemistry-climate hindcast model GFDL-AM3, which Lin et al. (2017) used to evaluate the increasing anthropogenic emissions in Asia, and Xu et al. (2018) used to examine the impact of direct tropospheric ozone transport at the Waliguan Station. Stratosphere-to-Troposphere Exchange Flux (STEFLUX, Putero et al., 2016) is a novel tool to quickly obtain

reliable identification of STT events occurring at a specific location and during a specified time window. STEFLUX relies on a compiled stratosphere-to-troposphere exchange climatology, making use of the ERA-Interim reanalysis dataset from the ECMWF, and a refined version of a well-established Lagrangian methodology. STEFLUX is able to detect stratospheric intrusion events on a regional scale, and it has the advantage of retaining additional information concerning the pathway of stratosphere-affected air masses, such as the location of tropopause crossing and other meteorological parameters along the

trajectories.

We applied STEFLUX to assess the possible contribution of STT to near-surface $O_3$ variability in the DA region (i.e., STEFLUX "target box", for further details on the methodology see Putero et al., 2016), and for identifying the measurement periods possibly affected by "deep" STT events (i.e., stratospheric air masses transferred down to the lower troposphere). For this work, we set the top lid of the box at 500 hPa, and the following geographical boundaries: 79–82 °S, and 76–79 °E.

A "deep" STT event at Kunlun Station was determined if at least 1 stratospheric trajectory crossed the 3-D target box.

4.3.2 Role of STT events at DA

The possible occurrence of stratospheric intrusion events, and their role in affecting the variability of near-surface $O_3$ and tropospheric air-chemistry in Antarctica has been investigated in several studies (Murayama et al., 1992; Roscoe, 2004; Stohl and Sodemann, 2010; Mihalikova and Kirkwood, 2013; Traversi et al., 2014; Traversi et al., 2017; Cristofanelli et al.,

2018). To provide a systematic assessment of the possible influence of "deep" STT events to the near-surface $O_3$ variability at Kunlun Station, we used the STEFLUX tool (see Sect. 4.3.1). Figure 12 shows the distribution of the occurrence of "deep" STT events over DA during the year. Although it is difficult to see a clear seasonal cycle, due to the low frequency of "deep" STT events, our results are in agreement with previous studies, indicating STT influence of up to 2% on a monthly basis (Stohl and Sodemann, 2010; Cristofanelli et al., 2018). According to our STEFLUX outputs, the highest frequency of "deep"

STT events was observed in May and August (1.1%). The frequency of occurrence of "deep" STT events identified by STEFLUX at Kunlun Station is about one order of magnitude lower than the occurrence of OEEs. Thus, a direct link of STT with OEEs interannual variability is unlikely, as also reported for DC station (Cristofanelli et al., 2018). Nevertheless, STT events can be a source of nitrates for the Antarctic atmosphere through different processes, thus indirectly affecting near-surface $O_3$ concentrations and favouring the presence of OEEs (Traversi et al., 2014; 2017).



## 5 Summary


Based on the in-situ monitoring data during 2016 at DA, the variation, formation, and decay mechanisms of near-surface $O_3$ were studied and compared with those at SP and ZS stations. The annual mean concentrations of near-surface $O_3$ at the DA, SP and ZS sites were 29.2 ± 7.5 ppb, 29.9 ± 5.0 ppb, and 24.1 ± 5.8 ppb, respectively. The near-surface $O_3$ concentrations were clearly higher in winter/polar night, with small fluctuations, than in the other seasons, which is different from the

patterns observed at low latitudes. The $O_3$ in inland areas was also higher than over the coast.

The diurnal variations showed nonsignificant regular patterns, and the range of the average diurnal concentration fluctuation was less than 1 ppb at all three stations. These findings suggest that the synoptic transport somehow controls the overall $O_3$ variability, as has been shown at the Amundsen-Scott and DC stations (Neff et al., 2008b; Cristofanelli et al., 2018).

At Kunlun station, it is unlikely that there is a direct relationship between STT and OEEs. The frequency of deep STT events

identified by STEFLUX is about an order of magnitude lower than OEEs, and reaches its highest frequency (1.1%) in May and August. As deduced by the STEFLUX application, "deep" STT events play a marginal role in steering the occurrence of OEEs at DA via "direct" transport of $O_3$ from the stratosphere/the free troposphere, to the surface. As explained in Cristofanelli et al. (2018), this can be related to an underestimation of STT "young" (i.e., < 4-day old) events by STEFLUX, or to insufficient spatial and vertical resolution from ERA-Interim to fully resolve the complex STT transport in the

Antarctic atmosphere (Mihalikova and Kirkwood, 2013). Despite this, STT can still represent a source of nitrates for the Antarctic snowpack, thus possibly affecting summer photochemical $O_3$ production. Therefore, it is important to carry out further studies to better assess these processes.

The characteristics and mechanisms of near-surface $O_3$ revealed in this paper have important implications for better understanding the formation and decay processes of near-surface $O_3$ in Antarctica, especially over the plateau areas.

Nevertheless, the lack of observations restricted our ability to amass more information. Long-term sustained observations at Dome A, Dome C, Dome F, SP, Vostok, and other locations, would greatly help in the future. In addition, the atmospheric chemical models are also valuable (Lin et al., 2017; Xu et al., 2018). In the future, we will compare and analyze different atmospheric chemical models and methods to obtain a more accurate analysis of the OEEs in Antarctica.

## 6 Data availability

All data presented in this paper are available in https://doi.org/10.5281/zenodo.3923517 (Ding et al., 2020). The data set covers the hourly average concentrations of near-surface ozone at three stations (i.e. SP, ZS, DA).



**Author contribution**

Minghu Ding and Biao Tian designed the experiments and wrote the manuscript; Minghu Ding carried out the experiments; Biao Tian analyzed the experimental results. Minghu Ding, Biao Tian and Davide Putero revised the manuscript; Davide

Putero run the STEFLUX tool. Michael Ashley, Zhenxi Zhu, Lifan Wang, Shihai Yang, Jie Tang, Chuanjin Li, Cunde Xiao and discussed the results.

**Acknowledgements:**

This work is financially supported by the National Natural Science Foundation of China (41771064), the Strategic Priority Research Program of Chinese Academy of Sciences (XDA20100300) and the Basic Fund of the Chinese Academy of

Meteorological Sciences (2018Z001). The observations were carried out by during the Chinese National Antarctic Research Expedition at the Zhongshan Station and the Kunlun Station. We are also grateful to NOAA for providing the HYSPLIT model and GFS meteorological files. Yaqiang Wang is the developer of MeteoInfo and provided generous help for the paper. PLATO-A was supported by the Australian Antarctic Division and with NCRIS funding through Astronomy Australia Limited.

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





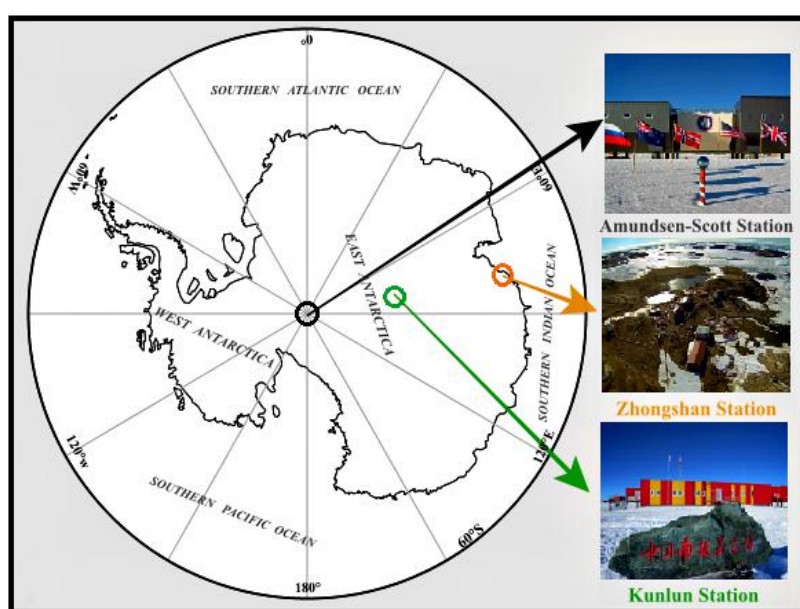

**Figure 1: Amundsen-Scott Station (South Pole, SP), Kunlun Station (Dome A, DA) and Zhongshan Station (ZS)**
**locations in Antarctica.**





**Table 1. The specifications of Model 205**

| Instrument performance | Model 205 |
|---|---|
| Measuring Range | 0ppb-100ppm |
| Weight (lb) | 4.7 lb |
| Working flow (l) | > 1.2 l |
| Data storage (lines) | 14336 |
| Working temperature (°C) | 0-50 |
| Indication error (ppb / d) | < 1 ppb/d |
| Response time (s) | 4 |
| Signal interface | RS232 |


**Table 2. The calibration record of ozone monitor**

| Date | Span Point (ppb) | Thermo 49ips (ppb) | Model 205  (ppb) |
|---|---|---|---|
| | 0 | -0.79 | 0.26 |
| | 20 | 19.99 | 20.73 |
| | 35 | 34.99 | 35.35 |
| 2015/10/5 | 50 | 50.02 | 50.73 |
| | 65 | 64.96 | 65.71 |
| | 80 | 79.99 | 80.48 |
| | 100 | 99.99 | 100.43 |
| | 120 | 119.96 | 120.31 |
| | 0 | -0.71 | 0.51 |
| | 20 | 20.00 | 21.68 |
| | 35 | 34.95 | 36.95 |
| 2017/5/6 | 50 | 50.01 | 52.17 |
| | 65 | 64.98 | 67.37 |
| | 80 | 79.99 | 82.88 |
| | 100 | 100.00 | 103.00 |
| | 120 | 119.92 | 124.10 |

**Table 3. Stability test of ozone monitor.**



| Time | Slope | Standard Uncertainty | Intercept | Standard Uncertainty |
|---|---|---|---|---|
| 2015/10/5 | 0.99936 | 0.00195 | 0.53861 | 0.13672 |
| 2017/5/6 | 1.02520 | 0.00264 | 0.85220 | 0.18491 |
| Average | 1.01228 | 0.00230 | 0.69541 | 0.16082 |
| Standard Error | 0.01827 | 0.00049 | 0.22174 | 0.03408 |


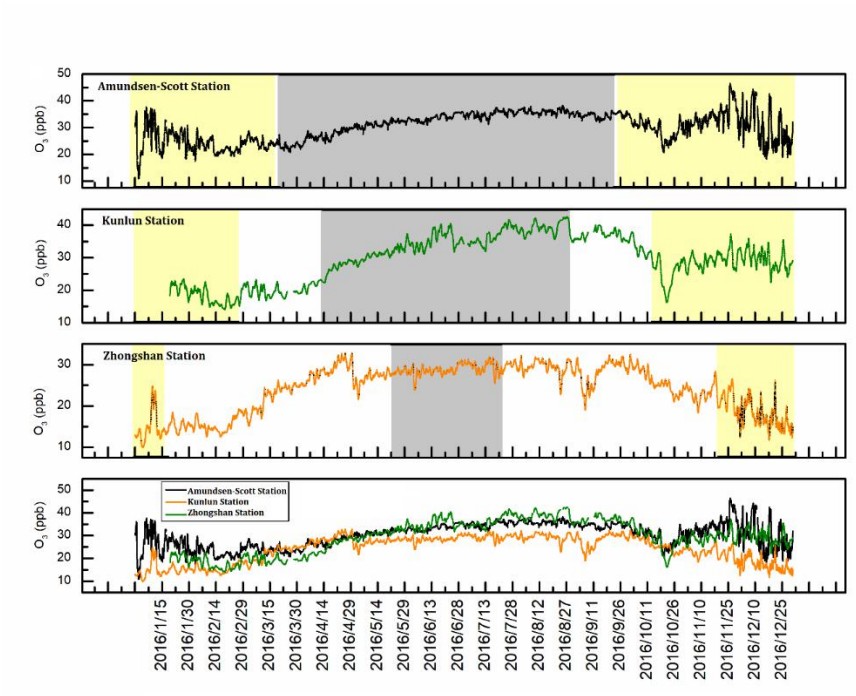

**Figure 2: Time series of near-surface $O_3$ at the SP, DA and ZS during 2016. Yellow (grey) shading identifies polar day**

**(night).**





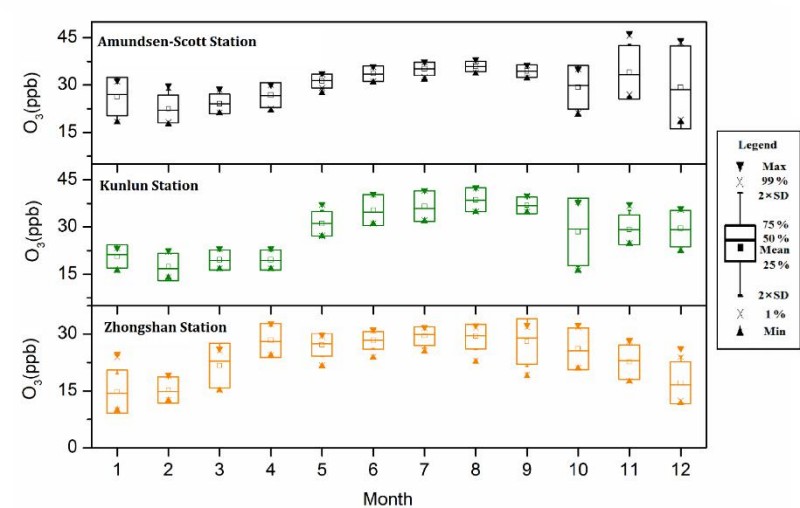

**Figure 3: Monthly average and statistical parameters of near-surface O₃ at the SP, DA and ZS during 2016.**

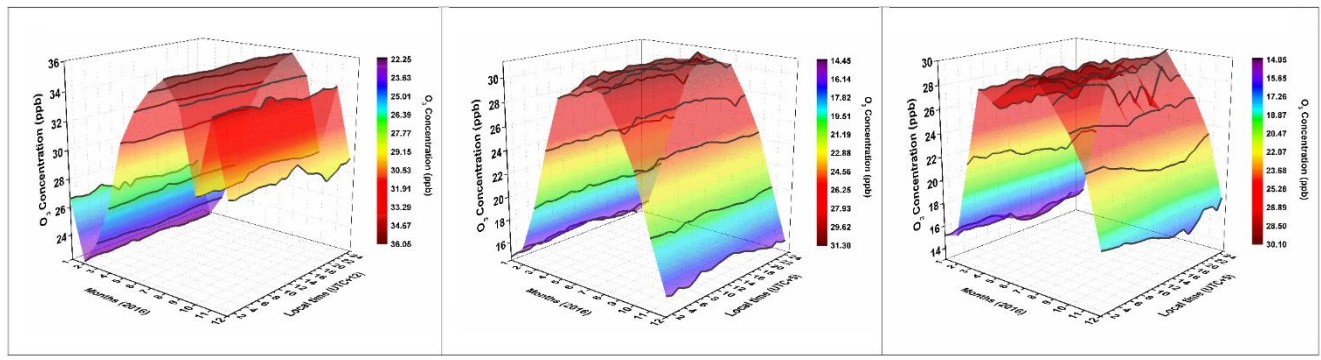

**Figure 4: Mean diurnal variations in near-surface O₃ concentrations at the SP (a), DA (b) and ZS (c) during 2016.**



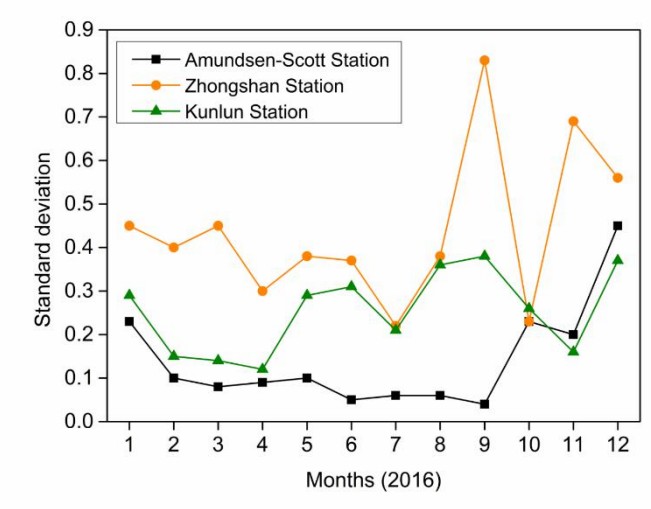


**Figure 5: Standard deviation of mean diurnal variations in near-surface O$_3$ concentrations at the SP, DA and ZS during 2016.**

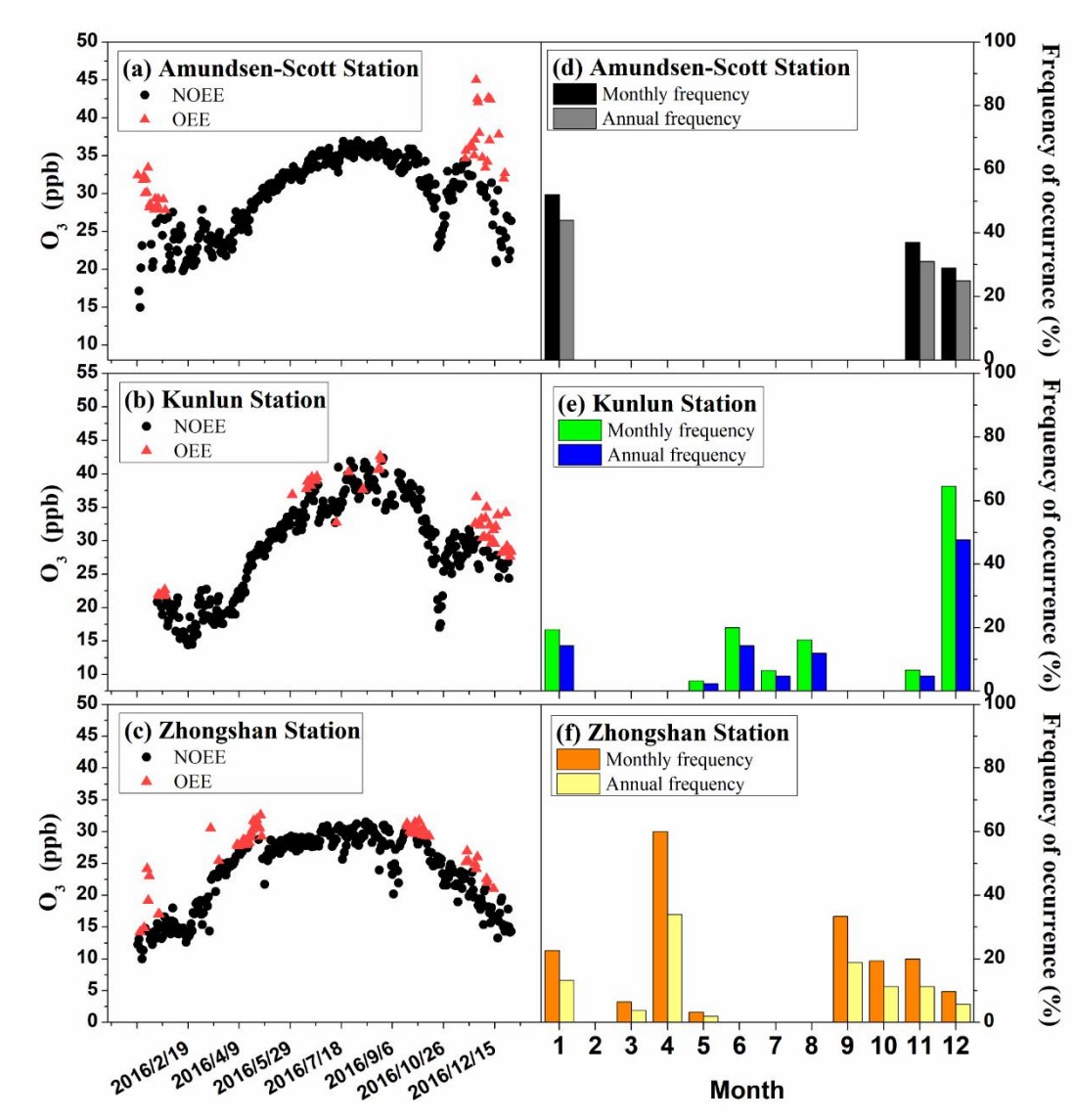

**Figure 6: (a, b and c) The OEEs and (d, e and f) averaged distribution of OEE occurrence among the different**

**months of 2016 at the three stations.** $Monthly\ frequency = \dfrac{\text{number of OEE days for each month}}{\text{number of days in the month}}$; $Annual\ frequency =$

$\dfrac{\text{number of OEE days for each month}}{\text{total number of OEE days}}$.



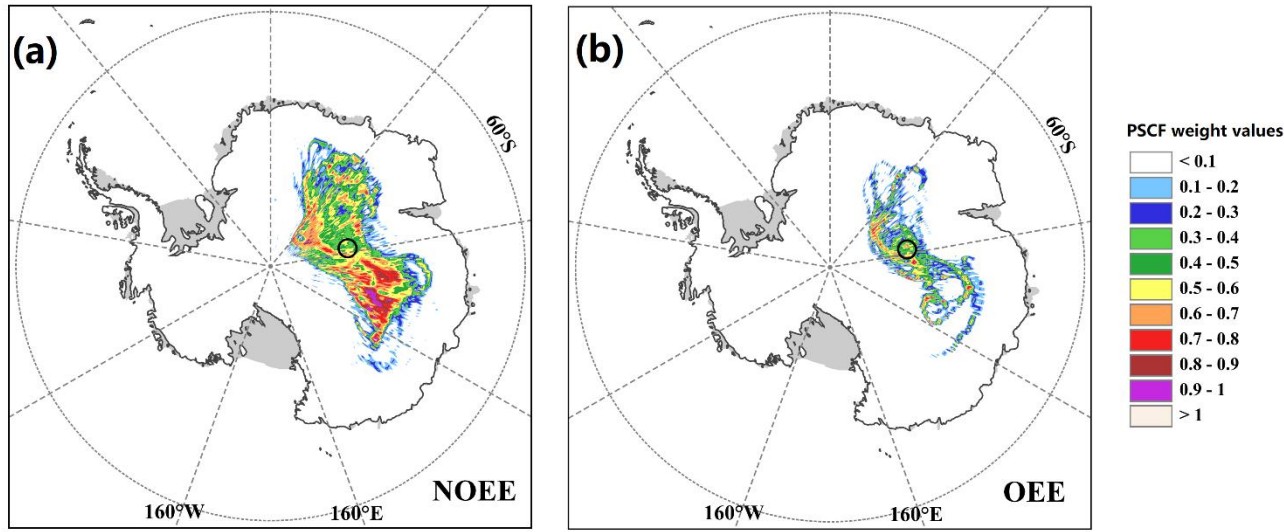

**Figure 7: Likely source areas of surface $O_3$ at Kunlun Station during the NOEE (a) and OEE (b) identified using the PSCF (Potential Source Contribution Function).**




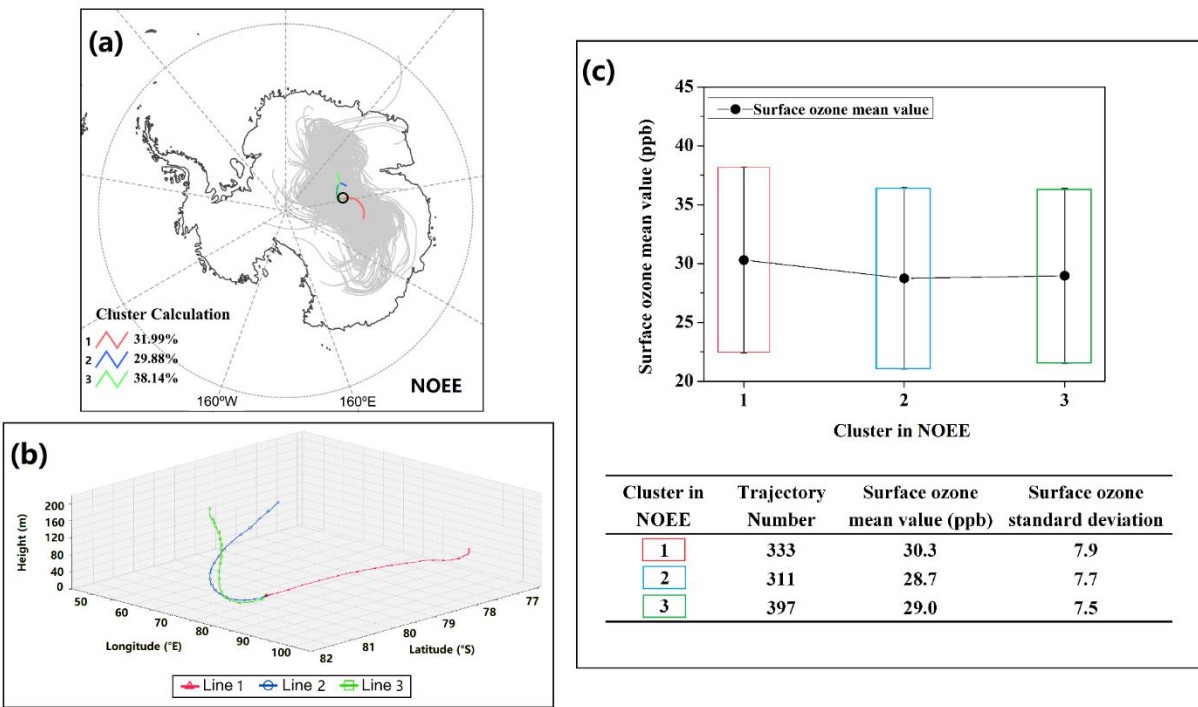

**Figure 8: Backward HYSPLIT trajectories for each measurement day (gray lines in Fig. 8a), and mean back trajectory for 3 HYSPLIT clusters (colored lines in Fig. 8a, 3D view shown in Fig. 8b) arriving at Kunlun Station during NOEEs. Subplot (c) shows the range of surface ozone mixing ratios measured at Kunlun Station by cluster.**




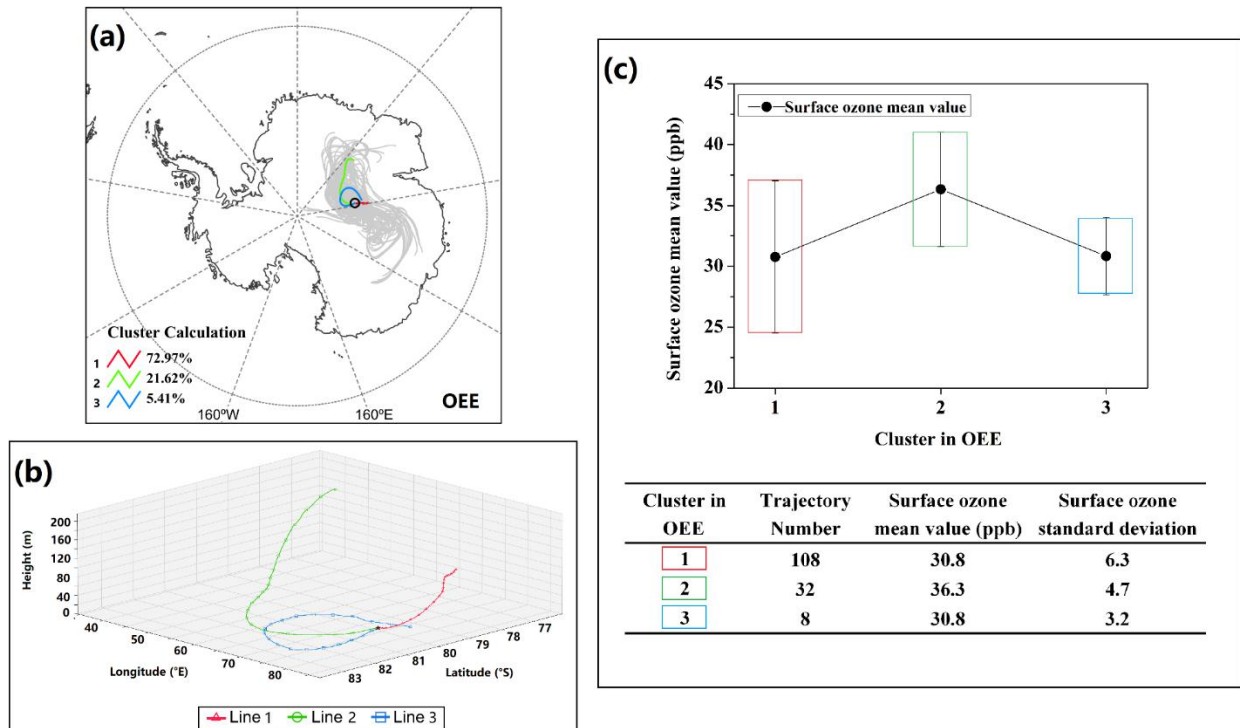

**Figure 9: Same as Fig. 8, but for OEEs.**

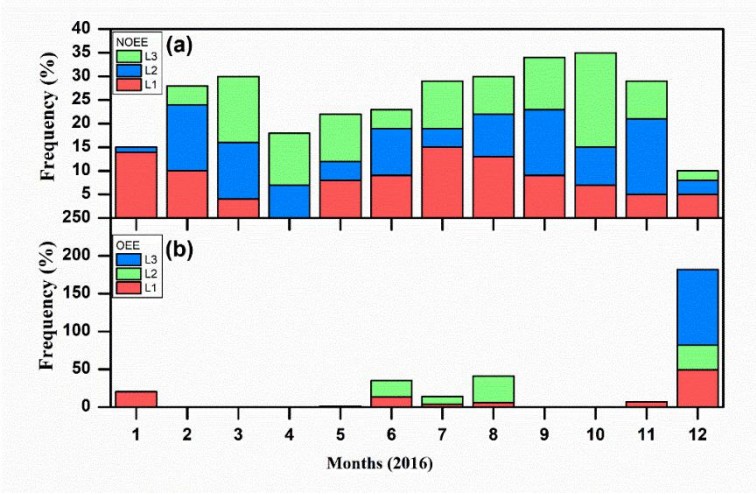

**Figure 10: Monthly frequency distribution of clustering trajectories (Line 1, 2, 3) during NOEEs and OEEs.**





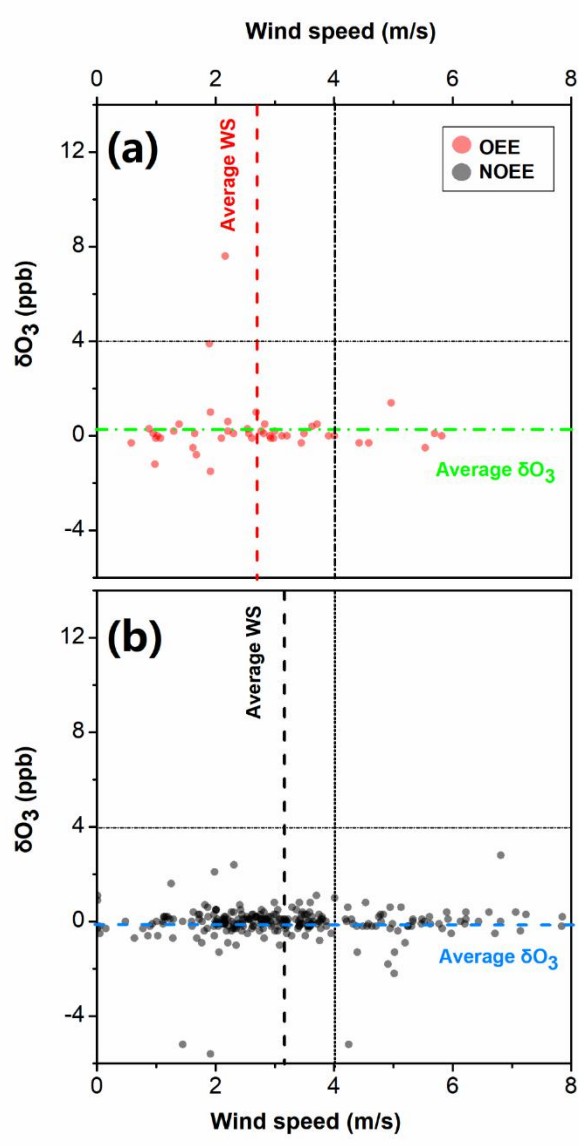

**Figure 11: Wind speed and $\delta O_3$ statistical distribution around OEEs (red dots) and NOEEs (black dots) at DA. Here,**

**$\delta O_3$ represents the growth rate of near-surface $O_3$ concentration, calculated by equation:**

$$\delta O_3 = \frac{\text{The O3 concentration at } T_n - \text{ The O3 concentration at } T_{n-1}}{\text{Time difference of } T_n \text{ and } T_{n-1}}$$



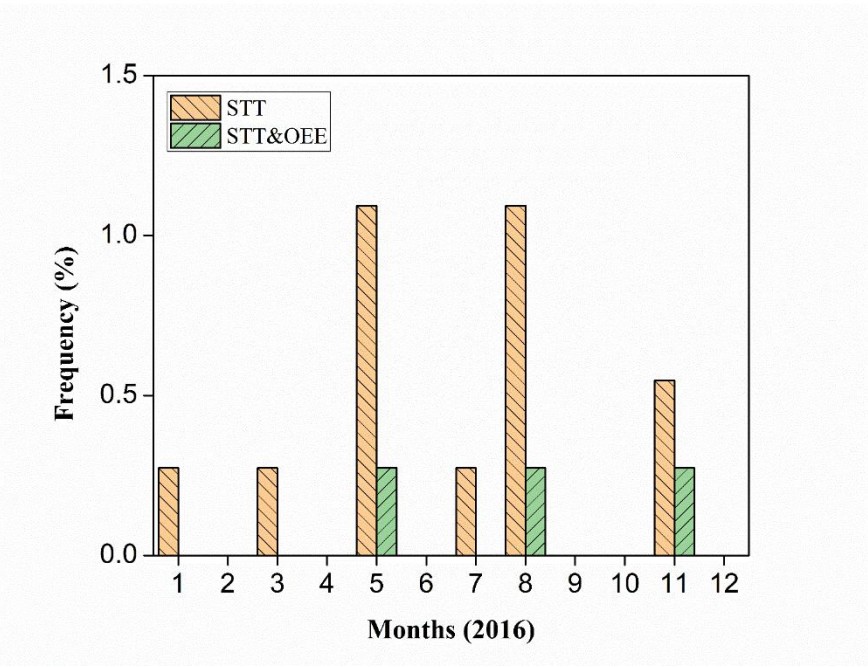


**Figure 12: Annual variation of "deep" STT events at Kunlun Station and the annual variation of it occurred at the same time with OEE over the period 2016, obtained by STEFLUX.**
