# Peer review of "Year-round record of near-surface ozone and "O3 Enhancement"

_Earth System Science Data, 2020_

## Referee Comment (RC1) · Anonymous Referee #1 · 29 Aug 2020

The manuscript is within the scope of ESSD. It presents scientifically significant material based on surface ozone measurements at three Antarctic stations. Of especial imporÂňtance are data of measurements at Dome A, the highest Antarctic plateau (ãĂIJ4000 m above sea level), which is one of the remotest areas on earth. The analysis of the data is reasonable and reliable, the data is unique. The authors should consider the following comments prior to publication.

General comments

1. I would appreciate it if the authors could please introduce more about the details of instruments and the measurements.

2. Because the results of trajectory clustering analysis are important in the discussion, there should be more description of trajectory clustering method.

3. L64-70. The specific chemical reaction process of nitrate aerosol photodegradation on snow pack should be increased. It is necessary to clarify the effect of NOx released by photodegradation on O3 emission from snow pack if it is possible.

4. In Section 3.3, the author's statement is too simple and arbitrary. The standard deviation of the average daily concentration in Zhongshan station was significantly higher than that in the other two inland stations. L259-L264 completely said that every solar chemical reaction had little effect on the concentration variation characteristics of the three stations, which was not rigorous. This paper focuses on the influence of daily photochemical reaction on the concentration variation characteristics of Dome A, and the difference of average daily concentration fluctuation between coastal stations and inland stations also needs to be discussed briefly. For example, coastal stations are easily affected by halogen gas mass in summer, and ODE (Ozone Depletion Event) is triggered (A.E. Jones et al., 2009 ), which has obvious impact on the fluctuation of average daily concentration in summer. Section 3.3 needs to add relevant references to support the author's statement.

5. In Section 4, the influence of STT on OEE is discussed by STEFLUX. It is also mentioned that STT can be judged by atmospheric chemical model (such as GFDL-AM3 and CAM-CHEM). Can author try to use CAM-CHEM Model to analyze the STT events in Dome A and compare the results with the OEE. At the same time, results of the two methods may be compared if it is possible.

6. Table 1 is not necessary, I suggest to delete it or move it to supplementary.

7. Fig. or Figure, please use the unified one in the whole paper.

Specific comments 1. Line 22, what is DA.

2. Line 232, only "in this part"? Sept→Sep?

[Figure]

3. Line 217-220, "concentration, molar ratio, mixting ratio", please be consistent with each others.

4. Line 282, from the results above, "it can be seen that" SP was characterized. . ..

5. Line 301, the wind of DA were "predominantly" from north and west. Prevalent may be better.

6. Line 317, have → has.

7. Line 361, As the station name has been abbraviatted, such as Amundsen-Scott →SP, all station names should be checked and properly used.

8. Line 375-376, the last sentence should be rewrite or removed.

9. I would suggest improving the readability of the label in Figure 4 if it is possible. It seems to not clear on my copy.

10. Figure 5, Standard deviations of mean diurnal variation in near-surface. . ..

11. Figure 8, what are the error bars.

12. Figure 11, $\delta$should be $\Delta$?

---

## Referee Comment (RC2) · Anonymous Referee #2 · 4 Sep 2020

Ozone is a major short-lived air pollutant when presents near ground, besides, it is a greenhouse gas that exerts direct influence on radiative forcing. The understanding of the variability and source of ground ozone in Antarctica remained limited, particularly in the inner Antarctica. In this paper, authors reported year-round observation of ozone in Dome A, the highest plateau in the Antarctica, they also complied observation data from South Pole and a costal site to make comparison. They revealed the occurrence of ozone enhancement events (OEEs) at Dome A and analyzed the possible sources and transport that contribute to the OEEs. The technical quality of the paper, including its observation and data analysis, is generally good. I have two major concerns on the manuscript.

[Figure]

1. The ESSD journal concentrates on datasets and the related process of data production. The current version of the paper did provide valuable time series of year-round ozone observation at Dome A, but it reads more like a research paper and author performed comprehensive diagnoses on the OEEs. I would leave the decision on the suitability of the paper for the journal to the editor.

2. Authors focused on ozone variability and OEEs at Dome A, they also included data from South Pole and a coastal site of Zhongshan Station and revealed different patterns of ozone variabilities in the three sites. However, in section 4, authors only analyzed the OEEs at Dome A site. The question is what's the purpose of including data from other two sites? Section 3 and 4 are not closely linked and I suggest authors to rethink the aims of the paper to stick to the main topic, e.g., differences in variabilities of ozone at three sites and possible reasons, or alternatively, the finding of strong OEEs at Dome A and its possible underlying mechanisms.

3. Figure 8 and 9 can be combined into one figure and the layouts of the figure should be re-designed to make it neat and clear.

---

## Author Response (AR1)

Dear Editor,

Thanks for your help and the two reviewers' kind suggestions. Accordingly, we have finished the revision of the manuscript, including (1) instrumental and field instruction in detail, (2) explanation on greater fluctuation of spring ozone at Zhonshan Station. We also analyzed the STT by CAM-Chem model but not added this part into manuscript, which can be found in the Reply to reviewers.

The new version is now ready for your consideration, we think. Please contact us if there is still flaws.

Best Regards,

Minghu
on behalf of all authors

**Reply for Anonymous Referee #1**

**The manuscript is within the scope of ESSD. It presents scientifically significant material based on surface ozone measurements at three Antarctic stations. Of especial importance are data of measurements at Dome A, the highest Antarctic plateau (4000 m above sea level), which is one of the remotest areas on earth. The analysis of the data is reasonable and reliable, the data is unique. The authors should consider the following comments prior to publication.**

General comments

1. I would appreciate it if the authors could please introduce more about the details of instruments and the measurements.

**Reply**:The detailed information of these instrument and observation introduction has been added in Section 2.1. Please find it in the Tracking version of the manuscript.

2. Because the results of trajectory clustering analysis are important in the discussion, there should be more description of trajectory clustering method.

**Reply**: The introduction of the trajectory cluster was added in Section 2.3, as bellow.

*"The air mass trajectories were assigned to distinct clusters according to their moving speed and direction using a k-means clustering algorithm (Wong, 1979). Concerning with this study focused on transport pathway of $O_3$, the clustering result with the smallest number was selected as done by Wang et al., (2014). It was found three clusters performance best to represented the meteorological characteristics of the transport pathways at DA. This number was then selected as the expected number of air mass trajectory clusters. A more detailed clustering procedure using the k-means algorithm can be found in Wang et al. (2014)."*

3. L64-70. The specific chemical reaction process of nitrate aerosol photodegradation on snow pack should be increased. It is necessary to clarify the effect of NOx released by photodegradation on $O_3$ emission from snow pack if it is possible.

**Reply**: According to your suggestion, the photochemical reaction process of snow surface is supplemented in Section 1. These explanation and correction have been added in the context (line 67-69).

*L67-L69: "As the solar irradiance and the nitrate aerosol concentration increase, the emission of $NO_X$ will increase through the photodenitrification process of the summer snowpack (e.g. $NO^{3-}+hv \rightarrow NO^{2}+O^{-}$; $O^{-}+H^{+} \rightarrow OH$; Honrath et al., 2000; Warneck et al., 1989)."*

4. In Section 3.3, the author's statement is too simple and arbitrary. The standard deviation of the average daily concentration in Zhongshan station was significantly higher than that in the other two inland stations. L259-L264 completely said that every solar chemical reaction had little effect on the concentration variation characteristics of the three stations, which was not rigorous. This paper focuses on the influence of daily photochemical reaction on the concentration variation characteristics of Dome A, and the difference of average daily concentration fluctuation between coastal stations and inland stations also needs to be

discussed briefly. For example, coastal stations are easily affected by halogen gas mass in summer, and ODE (Ozone Depletion Event) is triggered (A.E. Jones et al., 2009 ), which has obvious impact on the fluctuation of average daily concentration in summer. Section 3.3 needs to add relevant references to support the author's statement.

**Reply:** Thanks for your suggestion. As a coastal station, the average daily concentration fluctuation in Zhongshan station was obviously different with the two inland stations, which can be attributed into their background climates. In Spring, ODEs occur frequently at Zhongshan Station. And this phenomenon always accompanies with abrupt weather transit from continental dominant to oceanic dominant, in other words, the BrO brought by northerly wind from sea ice area could leaded to serious ozone depletion (Wang et al., 2011; Ye et al., 2018). Whereas at inland stations like DA and SP, there were rarely ODEs.

These explanation and correction have been added in the context.

5. In Section 4, the influence of STT on OEE is discussed by STEFLUX. It is also mentioned that STT can be judged by atmospheric chemical model (such as GFDL-AM3 and CAM-CHEM). Can author try to use CAM-CHEM Model to analyze the STT events in Dome A and compare the results with the OEE. At the same time, results of the two methods may be compared if it is possible.

**Reply:** Thank you for your suggestion. Starting to analyze the results of CAM-CHEM model after receiving this reply. But the results reflect an interesting phenomenon.

During the whole polar night, the results of CAM-CHEM model show that STT occur frequently over DA (Figure 1). However, the frequency of OEEs is lower than that of STT events. During the whole polar night, much times of STT promoted the increase of near surface ozone concentration of DA. However, on the one hand, the occurrence of OEE during the polar night is affected by STT. On the other hand, it may be related to the specific meteorological conditions. Based on the statistics of the meteorological elements of OEE and NOEE during the polar night (Table 1), the average wind speed was low and the average height of the planet boundary layer (PBL) was 66.46m. Moreover, the lower mean potential vorticity at 550 Hpa implies a stronger vertical downward transport process. The lower wind speed makes the high concentration ozone grow rapidly near the ground.

Compared with the analysis results of STEFLUX tool and CAM-CHEM model, the STEFLUX tool has a good selection for "deep" STT process. But the results show that "deep" STT process has little effect on OEE. However, the model results better reflect the frequent occurrence of STT over DA during the polar night, which is an important reason for the continuous accumulation of near surface ozone concentration during the polar night. However, the low frequency of OEE makes it difficult to establish a direct relationship with the model results. Under the frequent STT, the stable boundary layer condition with low wind speed near the ground is helpful to the occurrence of OEE.

Interestingly, on May 31, both the pattern results and STEFLUX results showed a strong STT, and OEE occurred on that day. The combination of the two methods makes us have more interesting findings. We hope to analyze such events in the future based on the meteorological data of DA and relevant model methods.

These findings is not the main purpose of the paper, we want to conduct a new detailed study on the comparison of the two methods in the future. However, if you insist to add this part, it can be done in the next modification.

[Figure]

Figure 1    The vertical distribution of ozone concentration over DA was calculated by CAM-CHEM model. During the polar night, the fluctuation of ozone concentration, the distribution of OEE and "deep" STT.

Table 1    The mean wind speed, air temperature, PBL and 550Hpa potential vorticity of OEEs and NOEEs during the polar night.

| Days | Wind Speed (m/s) | Temperature (℃) | PBL (m) | 550Hpa PV |
|---|---|---|---|---|
| OEEs | 2.77 | -33.07 | 66.46 | -3.47 |
| NOEEs | 3.13 | -35.66 | 32.29 | -2.55 |

6.  Table 1 is not necessary, I suggest to delete it or move it to supplementary.

**Reply**:According to your suggestion, the original Table 1 was deleted. The comparison of instrument parameters of the three stations is supplemented to replace the contents in Table 1.

7.  Fig. or Figure, please use the unified one in the whole paper.

**Reply**:It has been done.

**Specific comments**

1.  Line 22, what is DA.

**Reply:** It has been modified.

2. Line 232, only "in this part"? Sept—Sep?
**Reply:** It has been modified.

3. Line 217-220, "concentration, molar ratio, mixting ratio", please be consistent with each others.
**Reply:** The three words were unified as **concentration** in the full text.

4. Line 282, from the results above, "it can be seen that" SP was characterized....
**Reply:** It has been modified.

5. Line 301, the wind of DA were "predominantly" from north and west. Prevalent may be better.
**Reply:** It has been modified.

6. Line 317, have — has.
**Reply:** It has been modified.

7. Line 361, As the station name has been abbraviatted, such as Amundsen-Scott —SP, all station names should be checked and properly used.
**Reply:** It has been modified and the station names has been checked.

8. Line 375-376, the last sentence should be rewrite or removed.
**Reply:** The sentence has been removed.

9. I would suggest improving the readability of the label in Figure 4 if it is possible. It seems to not clear on my copy.
**Reply:** The figure in the PDF file were compressed, and the clear and non-destructive image was replaced.

[Figure]

10. Figure 5, Standard deviations of mean diurnal variation in near-surface....
**Reply:** It has been modified.

11. Figure 8, what are the error bars.

**Reply:** Error bars are the standard deviation of the same cluster. The explanation has been added in the caption.

12. Figure 11, $\delta$ should be $\triangle$?
**Reply:** It has been modified.

**Reply for Anonymous Referee #2**

**Ozone is a major short-lived air pollutant when presents near ground, besides, it is a greenhouse gas that exerts direct influence on radiative forcing. The understanding of the variability and source of ground ozone in Antarctica remained limited, particularly in the inner Antarctica. In this paper, authors reported year-round observation of ozone in Dome A, the highest plateau in the Antarctica, they also complied observation data from South Pole and a costal site to make comparison. They revealed the occurrence of ozone enhancement events (OEEs) at Dome A and analyzed the possible sources and transport that contribute to the OEEs. The technical quality of the paper, including its observation and data analysis, is generally good. I have two major concerns on the manuscript.**

General comments

1. The ESSD journal concentrates on datasets and the related process of data production. The current version of the paper did provide valuable time series of year-round ozone observation at Dome A, but it reads more like a research paper and author performed comprehensive diagnoses on the OEEs. I would leave the decision on the suitability of the paper for the journal to the editor.

**Reply**:Thank you for your advice. In this article, we did not only introduced the ozone data of DA, but also hoped to show some different characteristics with other Antarctic stations. It can provide more information to the readership.

To highlight the importance of the data and the reliability of the observation, detailed introduction on the three instruments and field plan were added in Section 2.1.

2. Authors focused on ozone variability and OEEs at Dome A, they also included data from South Pole and a coastal site of Zhongshan Station and revealed different patterns of ozone variabilities in the three sites. However, in section 4, authors only analyzed the OEEs at Dome A site. The question is what's the purpose of including data from other two sites? Section 3 and 4 are not closely linked and I suggest authors to rethink the aims of the paper to stick to the main topic, e.g., differences in variabilities of ozone at three sites and possible reasons, or alternatively, the finding of strong OEEs at Dome A and its possible underlying mechanisms.

**Reply**:Thank you for your comments. As you said, section 3 introduced the surface ozone characteristics of DA, SP and ZS, section 4 introduced OEEs only in DA but not SP and ZS. That is because we found only at DA, there were OEEs in winter. The other two, only have summer OEEs. And several studies have carried out on the causes of summer OEEs, such as Cristofanelli et al. (2018) and Legrand et al., (2016). They suggested continental transport was the key reason. Thus we wanted to focus on the unclear one, which was the winter OEEs.

In Section 4.1, the first paragraph introduced the method, the second paragraph introduced the general OEEs results of the three stations, the third paragraph introduced the differences of OEEs among the stations and highlighted the speciality of DA (winter OEEs). Then, the

fourth paragraph explained the findings of summer OEEs by previous studies. This logics was aimed to bring up the 4.2 and 4.3, which was the discussion on DA OEEs.

Anyway, if you think it is not suitable, we can delete this part and focus on general comparison among the three stations.

3. Figure 8 and 9 can be combined into one figure and the layouts of the figure should be re-designed to make it neat and clear.

**Reply:** The figures has been merged.

[Figure]

Figure 8: Backward HYSPLIT trajectories for each measurement day (gray lines in Figure 8a), and mean back trajectory for 3 HYSPLIT clusters (colored lines in Figure 8a, 3D view shown in Figure 8b) arriving at Kunlun Station during NOEEs. Subplot (c) shows the range of surface ozone concentrations measured at DA by cluster. Error bars are the standard deviation of the same cluster. Same as subplot (a, b, c), but subplot (d, e, f) for OEEs.

[revised manuscript text omitted]

**2 Sites and methods description**

**2.1 Near-surface ozone observations**

介绍测量 O3 的各种仪器，特别是三个站涉及到的仪器，及其优劣。DA, SP and ZS was used the Model 205 Dual Beam, Thremo 49C, and EC9810A and Model 205 ozone monitors, respectively. All three instruments are based on ultraviolet spectrophotometry to measure ozone concentration. The ultraviolet spectrophotometry is based on the law of Beer Lambert. The method of measuring ozone concentration by detecting the maximum absorption value of ozone at the wavelength of 253.7 nm. The principles are as follows:

$$I = I_0 \, exp(-\alpha L C) \qquad\qquad (1)$$

The $I$ I is the light intensity after the airflow passes through. The $I_0$ Io is the light intensity when the airflow does not pass through. The $\alpha$ a is the absorption coefficient. The $L$ L is the absorption path, and the $C$ C is the ozone concentration in the absorption gas.

Table 1 shows that there are great differences in measuring range, weight, working flow and data storage. Clearly, the Model 205 ozone monitor is smaller and portable, but the measuring range of the instrument is small and the data storage capacity is weak. These three monitors were made use of two detection cells to improve precision, baseline stability, and response time. In the dual beam instrument, UV light intensity measurements $I$ (O₃-scrubbed air) and $I_0$ (unscrubbed air) are performed simultaneously. Combined with other improvements, The Model 205 Dual Beam Ozone Monitor is the fastest UV-based O₃ monitor on the market, with such small size, weight, and power requirements characteristics. Fast measurements are particularly desirable for unattended stations, aircraft and balloon measurements, where high spatial resolution is desired. The Model 205 Dual Beam Ozone Monitor (205-2B) is an Environmental Protection Agency (EPA) federal equivalent method (FEM).

There are manyseveral methods to measure the concentration of ozone, such asincluding ultraviolet spectrophotometry, iodometry, sodium indigo disulfonate spectrophotometry, gas chromatography, chemiluminescence, fluorescence

spectrophotometry and long optical path differential absorption spectrometry, etc. (e.g. Wang et al.,2017). Of them the ultraviolet spectrophotometry is the most popular for the surface ozone monitoring  and applied into  many commercial instruments . The common ones such as Thermo 49C (Liu et al., 2006), API 400E (Sprovieri et al., 2003), ESA O342M (Lei et al., 2014), Ecotech 9810B (Moura et al., 2011), have been used in many regions for their larger measuring range and high precision, but they are expensive, need plenty of power supply and regular maintenance. Recently, more and more studies chosed the portable ozone monitors such as Model 205, Aeroqual Series 500, POM, due to its advantages of small volume, low price, low energy consumption and good applicability for field observation (e.g. Johnson et al., 2014; Lin et al., 2015; Sagona et al., 2018). In Antarctica, only few stations have carried out continuous ozone monitoring and all of them were equipped with the common types, that is Thermo/Ecotech types, as we know.

~~One is large-scale fixed monitoring instruments, mainly including Thermo 49C (Liu et al., 2006), API 400E (Sprovieri et al., 2003), ESA O342M (Lei et al., 2014), Ecotech 9810B (Moura et al., 2011), etc. This kind of instrument has large volume, high price, long measuring range, high precision, high energy consumption and needs regular maintenance. The other is portable monitor, mainly Model 205 (Johnson et al., 2014), Aeroqual Series 500 (Lin et al., 2015), POM (Sagona Jet al., 2018) etc. This kind of instrument has the advantages of small volume, low price, low energy consumption and good applicability for field observation.~~

~~At present, there are not many stations to carry out long-term near surface ozone observation in Antarctica. Because the operation of large fixed monitoring instruments requires continuous power supply and routine maintenance. Therefore, only a few countries' perennial stations (Syowa, Neumayer, Halley, Arrival Heights, South Pole, Zhongshan) have made continuous observations. The surface ozone data of some stations could be understood from the World Data Centre for Greenhouse Gases (WDCGG). However, there are few studies on the use of portable monitors for Antarctic ozone observation.~~

The Kunlun Station (80°25'02"S, 77°06'59"E, altitude 4087 m) is located in the DA area, on the summit of the east Antarctic Ice Sheet (Figure 1). The only continuous power supply is the PLATO-A observatory,  which can also provide internet access via the Iridium satellite network (detailed introduction of PLATO observatory can be found in Lawrence et al. (2009)). Due to the limitation of energy consumption and conditions encountered during transportation from coast to the Dome, larger monitors such as Thermo 49i  cannot be used. Thus,  on the 1st of Jan 2016, we deployed a Model 205 Dual Beam Ozone Monitor (205 2B) during the 33rd Chinese National Antarctic Research Expedition.

160 The instrument has been certified by Environmental Protection Agency and makes use of two detection cells to improve its precision, baseline stability, and response time. In the dual beam instrument, UV light intensity measurements $I_0$ (O$_3$-scrubbed air) and $I$ (unscrubbed air) are performed simultaneously (Wang et al., 2017). And it is the fastest UV-based O$_3$ monitor till now, with

165  a small size, light weight, and low power requirements  (Ssupplementary Table S1). Especially, quick response is particularly desirable for unattended stations, aircraft and balloon measurements In Dome A, we use Teflon pipeline to connect the free air at ~4m above surface with the instrument. At the inlet of the pipeline, a Thermo 47mm filter was used to prevent snow particles. During

170 the observation, the instrument was set at the sampling frequency of once an hour, and the data was transmitted to the observatory computer through RS232 and sent to Beijing by satellite.

175

The Zhongshan Station (69°22'12"S, 76°21'49"E, altitude 18.5 m) is located at the edge of the east Antarctic Ice Sheet (Figure  1 ). The atmospheric chemistry observatory was constructed at the Swan Ridge  northwest of the Nella fjord, at where we installed a UV absorption near-surface O3 analyzer (EC9810A) for long-term near-surface O$_3$ monitoring in Jan. 2008. The air inlet

180  was 4 m above the surface, and connected to the analyzer through the Teflon pipe The observational frequency was 3 min, and the data were transferred in real time to Beijing. Furthermore, to prevent data losses, a CR1000 data logger was used to record the data output in real time. Every three months, the O$_3$ analyzer was calibrated using the EC9811 O$_3$ calibrator, and 5 standard concentrations of O$_3$ gas were

185 generated for each calibration. The calibration concentration and measured concentration underwent correlation analysis, and seasonal calibration results were generated every three months. In 2016, 5 calibrations were made, and the appropriate correlation coefficients (r) were all greater than 0.9995.

The Amundsen-Scott Station (89°59'51.19 "S, 139°16'22.41" E, altitude 2835 m) is located at the SP and operated by the United States.  In 2016, an Thermo 49C ozone monitor was used and 5-minute

190 and 1-hour data was uploaded to GAW (Global Atmospheric Watch) every month.

1-hour frequency data every month. The near-surface O₃ datarecord used in this paper were was downloaded from the Earth System Research Laboratory Global Monitoring Division under the NOAA (https://www.esrl.noaa.gov/gmd/dv/data).

The hourly data of these stations collated here are available at https://doi.org/10.5281/zenodo.3923517 (Ding et al., 2020).

**2.2 Calibration process and results**

195 Generally, the zero point, span point and operation parameters of the $O_3$ monitor should be checked before each operation. The zero point should be checked regularly during continuous observation. While such regular calibration was done at the Global Atmosphere Watch (GAW) and Zhongshan Station, it was not possible at DA due to the lack of logistic support and the extreme environment. To minimize the error and evaluate the accuracy of the experiment, a UV-absorption $O_3$ calibrator Thermo 49i-PS was used to examine the Model 205. The calibration procedure follows China's environmental protection

200 standard "ambient air - Determination of ozone - ultraviolet method" (HJ590-2010) (http://www.mee.gov.cn/gkml/sthjbgw/sthjbgg/201808/t20180815_451411.htm) which is more strict than USEPA (Ref: USEPA . —Quality Assurance Handbook for Air Pollution Measurement Systems Volume II: Ambient Air Quality Monitoring Program[EB/OL].[2008-12-01]. http://www.epa.gov/ttn/amtic/files/ambient/pm25/qa/ QAHandbook-Vol-II.pdf.): the slope of calibration curve ranges between 0.95-1.05, and the intercept ranges between -5-5 ppb. Instruments used in the

205 calibration process include a DOA-p512-bn air compressor (USA), in addition to the Thermo 49ips $O_3$ calibrator and the Model 205 $O_3$ monitor. Before each test, the $O_3$ calibrator and the $O_3$ monitor were turned on and preheated for 12 hours, and the measuring range was set to 400 ppb. We first generate a zero concentration using the Thermo 49ips and, once the analyzer response has stabilized on zero reading, we adjusted the Model 205's internal zero setting to matches the zero air source. Then, $O_3$ airflow at 400 ppb level was generated and injected into the analyzer, and a correction factor was calculated

210 based on the observed value, which was then loaded into the Model 205 configuration.

After the calibration of the internal zero/span settings, a second stage of calibration was performed involving multi-point verification to check the response and stability of the analyzer. On Oct 5th 2015 (before the instrument was shipped) and May 6th 2017(the day that the instrument was transported back from Antarctica), a zero and 7 upscale points (0, 20, 35, 50, 65, 80, 100, 120 ppb) encompassing the full scale of the observation range (Table —S2121), were generated by the Thermo

215 49ips to test the Model 205 analyzer. Each point was observed for 15 min, during the last 10 minutes of which readings were taken every minute of the calibrator and analyzer. Based on this experiment, the slope and intercept of the calibration curve were calculated by least squares. The results are shown in Table 212, it can be concluded that the slopes of the linear correction curve were 0.99936 and 1.02520, and the intercepts were 0.53861 and 0.85220l (Table 3222), which fulfilled the requirements of HJ590-2010 and USEPA.

220 Another challenge when monitoring the atmosphere is the stability of the analyzer, which includes the analyzer's response time. Similarly with the regular calibration, it could not be performed during the observation period, but it was reassuring that the Model 205 was still in good condition when we did the multi-point verification in May 2017, as shown in Table

. The slope and intercept of the two calibration curves changed little and the standard uncertainties were small. To further test the stability, data consistency was also examined and the mean absolute deviation between two adjacent values was only 0.09 ppb. The largest difference was 0.61 ppb, indicating that the analyzer was stable and reliable.

Before analysis, a variance test was used to remove abnormal data based on the Laida criterion method, which assumes that the records obeyed a normal distribution. The formula is $\left| x_i - \overline{x} \right| > 3\sigma$, where $x_i$ is the measured value, $\overline{x}$ is the time series mean and $\sigma$ is the standard deviation. After processing, 99.3%, 99.6%, and 89.3% of the hourly mean data were retained from the Amundsen-Scott Station, Zhongshan Station and Kunlun Station, respectively.

**2.3 Air mass back-trajectory calculations**

Gridded meteorological data for backward trajectories in Hybrid Single-Particle Lagrangian Integrated Trajectory (HYSPLIT) were obtained from the Global Data Assimilation System (GDAS-1) operated by the U.S. National Oceanic and Atmospheric Administration (NOAA) with 1°×1° horizontal resolution and 23 vertical levels, from 1000 hPa to 20 hPa (http://www.arl.noaa.gov/gdas1.php).

The HYSPLIT backward air mass trajectory model was previously applied to atmospheric research in Antarctica (Legrand et al., 2009; Hara et al., 2011). We used the HYSPLIT model in this paper to analyse the impact of varying air mass sources and the intrusion of stratospheric $O_3$. Backward trajectories and clusters were calculated using the US National Oceanic and Atmospheric Administration (NOAA)-HYSPLIT model (Draxler and Rolph, 2003; http://ready.arl.noaa.gov/HYSPLIT.php), which is a free software plug-in for MeteoInfo (Wang, 2014; http://meteothink.org/). The backward trajectories starting height was set at 20 m above the surface and the total run times was 120 hours for each backward trajectory, and each run was performed in time intervals of 6 hours (00:00, 06:00, 12:00, 18:00).

The integral error part of the trajectory calculation error can be estimated by simulating the backward trajectory at the end of the forward trajectory and comparing the differences of the tracks. The starting point of the backward integration is set as (77.12 ° E, 80.42 °S, 20m a.g.l.), the backward integration is 120 hours. Then the point reached at this time is taken as the starting point, and a forward simulation is made for 120h. In this simulation experiment, the contribution of integration error to trajectory calculation error is very small within the first 72 hours. With the extension of integration time, the integration error slightly increases.

The air mass trajectories were assigned to distinct clusters according to their moving speed and direction using a k-means clustering algorithm (Wong, 1979). Concerning with this study focused on transport pathway of $O_3$, the clustering result with the smallest number was selected as done by (Wang et al. (2014). It was found three clusters performance best to represent the meteorological characteristics of the transport pathways at DA. This number was then selected as the expected number of air mass trajectory clusters. A more detailed clustering procedure using the k-means algorithm can be found in Wang et al. (2014).

[revised manuscript text omitted]

station area. However, due to the influence of polar vortices, ODEs of inland stations are not obvious (Wang Y et al., 2011; Ye et al., 2018)..Obviously, the average daily concentration fluctuation in Zhongshan station was obviously different with the two inland stations, which can be attributed into their background climates. In Spring, ODEs occur frequently at

340 Zhongshan Station. And this phenomenon always accompanies with abrupt weather transit from continental dominant to oceanic dominant, in other words, the BrO brought by northerly wind from sea ice area could leaded to serious ozone depletion (Wang Y et al., 2011; Ye et al., 2018). Whereas at inland stations like DA and SP, there were rarely ODEs.

On the whole, tThe mean diurnal variations in different time periods were not obvious, and the mean diurnal concentrations of the three stations fluctuated within a range of less than 1 ppb, indicating that daily photochemistry reactions were not the

345 dominant factor in near-surface O₃ at the three stations. The magnitude of the diurnal variation was low, which is similar to the variations found atof other Antarctic stations,,Neumayer Dome C and Marambio McMurdo for instance (Gruzdev et al., 1993; Ghude et al., 2005; NadzirOltmans et al., 20108).

**4 Ozone under OEEs at the Kunlun Station**

**4.1 Identification of OEEs**

350 Our method to select the days characterized by OEEs is based on the procedure used in Cristofanelli et al. (2018). First, a sinusoidal fit is used to calculate the O₃ annual cycle not affected by the OEEs, then a probability density function (PDF) of the deviations from the sinusoidal fit is calculated, with the application of a Gaussian fit to the obtained PDF. As reported in Giostra et al. (2011), the deviations from the Gaussian distribution (calculated by using the Origin 9© statistical tool) can be used to identify observations affected by non-background variability. We computed the further Gaussian fitting of PDF

355 points beyond 1 σ (standard deviation) of the Gaussian PDF, and determined the non-background O₃ daily values that may be affected by "anomalous" O₃ enhancement. The intersection of the two fitting curves is taken as our screening threshold (3.4 ppb at SP, 3.4 ppb at Da and 2.5 ppbzs at ZS). Figures 6a, 6b and 6c show OEE days and NOEE days at these three stations, while Figures 6d, 6e and 6f report the distribution frequency of OEE days.

In total, 42 days at DA were found to be affected by anomalous OEEs: 14.3% in January, 2.4% in May, 14.3% in June, 4.8%

360 in July, 11.9% in August, 4.8% in November and 47.6% in December (Figure 6e, blue bars). This result clearly indicates that half of the anomalous days occurred in December, followed by January and June. At SP, 36 days with OEEs were found in 2016: 44.4% in January, 30.6% in November, and 25% in December (Figure 6d, grey bars). Apparently, OEEs occur only in summertime at this measurement site. ZS was characterized by more days with OEEs: 53 days in April (34.0%), followed by September (18.9%), January (13.2%), October (11.3%), November (11.3%), December (5.7%) March (3.8%) and May

365 (1.9%) (Figure 6f, yellow bars).from the results above, "it can be seen that" SP was characterized....

From the results above, it can be seen that SP was characterized by concentrated OEE occurrences, and ZS had the most scattered OEEs pattern. In addition, all OEEs at SP and ZS occurred during the Antarctic warm season, and no OEEs were present during the polar night, similarly to the pattern observed at DC (Cristofanelli et al., 2018). In contrast, the OEEs also occurred during the polar night in DA, and the number of OEE occurrence days accounted for up to 33% of the total number of events throughout the year. This is the main reason of the large variations of daily average concentration,  during the polar night of DA .

[revised manuscript text omitted]

Moura, Bárbara Bâesso, Souza, Sílvia Ribeiro de, Alves E S: Respostas estruturais em Ipomoea nil (L.) Roth 'Scarlet O'Hara' (Convolvulaceae) exposta ao ozônio, Acta Botanica Brasilica, 25(1):122-129, https://doi.org/10.1590/S0102-33062011000100015, 2011.

Murayama, S., Nakazawa, T., Tanaka, M., Aoki, S., Kawaguchi, S.: Variations of tropospheric ozone concentration over
605 Syowa Station, Antarctica, Tellus B 44, 262–272, http://dx.doi.org/10.1034/j.1600-0889.1992.t01-3-00004.x, 1992.

Nadzir M S M , Ashfold M J , Khan M F , et al.: Spatial-temporal variations in surface ozone over Ushuaia and the Antarctic region: observations from in situ measurements, satellite data, and global models, Environmental ence & Pollution Research, http://dx.doi.org/10.1007/s11356-017-0521-1, 2018.

[revised manuscript text omitted]

Wong, JAHA.: Algorithm AS 136: A K-Means Clustering Algorithm, Journal of the Royal Statistical Society, 28:100-108, https://doi.org/10.2307/2346830, 1979.

Xu, W., Xu, X., Lin, M., Lin, W., Tarasick, D., Tang, J., Ma, J., and Zheng, X.: Long-term trends of surface ozone and its influencing factors at the Mt Waliguan GAW station, China – Part 2: The roles of anthropogenic emissions and climate variability, Atmos. Chem. Phys., 18, 773–798, https://doi.org/10.5194/acp-18-773-2018, 2018.

Ye, L., Bian, L., Tang, J., Ding, M., Zheng, X., Gao, Z.: A study on surface ozone depletion episodes over the A ntarctic coast, Acta Meteorologica Sinica, 75(3),506-516, https://doi.org/ 10.11676/qxxb2017.027, 2017.

Yin, X., Kang, S., de Foy, B., Cong, Z., Luo, J., Zhang, L., Ma, Y., Zhang, G., Rupakheti, D., and Zhang, Q.: Surface ozone at Nam Co in the inland Tibetan Plateau: variation, synthesis comparison and regional representativeness, Atmos. Chem. Phys., 17, 11293-11311, https://doi.org/10.5194/acp-17-11293-2017, 2017.

675

680

685

690

695

[Figure]

**Figure 1: Amundsen-Scott Station (South Pole, SP), Kunlun Station (Dome A, DA) and Zhongshan Station (ZS) locations in Antarctica.**

Table 1. The specifications of Model 205

| Instrument performance | Model 205 |
| --- | --- |
| Measuring Range | 0ppb-100ppm |

| | |
|---|---|
| Weight (lb) | 4.7 lb |
| Working flow (l) | >1.2 l |
| Data storage (lines) | 14336 |
| Working temperature (°C) | 0-50 |
| Indication error (ppb / d) | <1 ppb/d |
| Response time (s) | 4 |
| Signal interface | RS232 |

710 **Table.1 Comparison of the working parameters in the three instruments**

| Instrument performance | Model 205 | Ecotech 9810A | Thermo 49C |
|---|---|---|---|
| Measuring Range | 0ppb-100ppm | 0ppb-20ppm | 0ppb-200ppm |
| Weight (kg) | 2.2kg | 21 | 15.9 |
| Working flow (L/min) | >1.2 | $0.5*10^{-3}$ | 1~3 |
| Data storage (lines) | 14336 | 50400 | 115200 |
| Working temperature (°C) | 0-50 | 5-40 | 0-45 |
| Indication error (ppb / d) | <1 ppb/d | <1 ppb/d | <1 ppb/d |
| Response time (s) | 4 | 60 | 20 |
| Signal interface | RS232 | RS232, USB | RS232, RS485,RJ45 |

**Table 212. The calibration record of ozone monitor**

| Date | Span Point (ppb) | Thermo 49ips (ppb) | Model 205 (ppb) |
|---|---|---|---|
| | 0 | -0.79 | 0.26 |
| | 20 | 19.99 | 20.73 |
| 2015/10/5 | 35 | 34.99 | 35.35 |
| | 50 | 50.02 | 50.73 |
| | 65 | 64.96 | 65.71 |

| | 80 | 79.99 | 80.48 |
| | 100 | 99.99 | 100.43 |
| | 120 | 119.96 | 120.31 |
| | 0 | -0.71 | 0.51 |
| | 20 | 20.00 | 21.68 |
| | 35 | 34.95 | 36.95 |
| | 50 | 50.01 | 52.17 |
| 2017/5/6 | 65 | 64.98 | 67.37 |
| | 80 | 79.99 | 82.88 |
| | 100 | 100.00 | 103.00 |
| | 120 | 119.92 | 124.10 |

**Table 1. The calibration record of ozone monitor**

| Date | Span Point (ppb) | Thermo 49ips (ppb) | Model 205 (ppb) |
|---|---|---|---|
| | 0 | -0.79 | 0.26 |
| | 20 | 19.99 | 20.73 |
| | 35 | 34.99 | 35.35 |
| | 50 | 50.02 | 50.73 |
| 2015/10/5 | 65 | 64.96 | 65.71 |
| | 80 | 79.99 | 80.48 |
| | 100 | 99.99 | 100.43 |
| | 120 | 119.96 | 120.31 |
| | 0 | -0.71 | 0.51 |
| | 20 | 20.00 | 21.68 |
| | 35 | 34.95 | 36.95 |
| | 50 | 50.01 | 52.17 |
| 2017/5/6 | 65 | 64.98 | 67.37 |
| | 80 | 79.99 | 82.88 |
| | 100 | 100.00 | 103.00 |
| | 120 | 119.92 | 124.10 |

715

720

**Table 32. Stability test of ozone monitor.**

| Time | Slope | Standard Uncertainty | Intercept | Standard Uncertainty |
|---|---|---|---|---|
| 2015/10/5 | 0.99936 | 0.00195 | 0.53861 | 0.13672 |
| 2017/5/6 | 1.02520 | 0.00264 | 0.85220 | 0.18491 |
| Average | 1.01228 | 0.00230 | 0.69541 | 0.16082 |
| Standard Error | 0.01827 | 0.00049 | 0.22174 | 0.03408 |

[Figure]

[Figure]

**Figure 2: Time series of near-surface** O$_3$ **at the SP, DA and ZS during 2016. Yellow (grey) shading identifies polar day (night).**

[Figure]

**Figure 3: Monthly average and statistical parameters of near-surface O₃ at the SP, DA and ZS during 2016.**

[Figure]

735

**Figure 4: Mean diurnal variations in near-surface O₃ concentrations at the SP (a), DA (b) and ZS (c) during 2016.**

[Figure]

**Figure 5:** Standard deviations of mean diurnal variation in near-surface O$_3$ concentrations at the SP, DA and ZS during 2016.

[Figure]

[Figure]

[Figure]

**Figure 6: (a, b and c) The OEEs and (d, e and f) averaged distribution of OEE occurrence among the different**

745 **months of 2016 at the three stations.** $\textit{Monthly frequency} = \frac{\text{number of OEE days for each month}}{\text{number of days in the month}}$; $\textit{Annual frequency} =$

$\frac{\text{number of OEE days for each month}}{\text{total number of OEE days}}$.

[Figure]

**Figure 7: Likely source areas of surface O$_3$ at Kunlun Station during the NOEE (a) and OEE (b) identified using the**
750   **PSCF (Potential Source Contribution Function).**

[Figure]

[Figure]

[Figure]

755  **Figure 8: Backward HYSPLIT trajectories for each measurement day (gray lines in** **Figure 8a), and mean back trajectory for 3 HYSPLIT clusters (colored lines in** **Figure 8a, 3D view shown in** **Figure 8b) arriving at Kunlun Station during NOEEs. Subplot (c) shows the range of surface ozone** **concentrations measured at** **DA by cluster. Error bar are the standard deviation of the same cluster. Same as subplot (a, b, c), but subplot (d, e, f) for OEEs.**

760

[Figure]

[Figure]

[Figure]

**Figure 9: Same as Fig.Figure 8, but for OEEs.**

[Figure]

765

**Figure 9: Monthly frequency distribution of clustering trajectories (Line 1, 2, 3) during NOEEs and OEEs.**

[Figure]

[Figure]

Figure 1110: Wind speed and δΔO₃ O₃ statistical distribution around OEEs (red dots) and NOEEs (black dots) at DA in polar night. Here, ΔO₃δO3 represents the growth rate of near-surface O₃ concentration, calculated by equation:

$$\Delta O3\delta O_3 = \frac{\text{The O3 concentration at } T_n - \text{ The O3 concentration at } T_{n-1}}{\text{Time difference of } T_n \text{ and } T_{n-1}}$$

[Figure]

[Figure]

775  **Figure 11: Annual variation of "deep" STT events at Kunlun Station and the annual variation of it occurred at the same time with OEE over the period 2016, obtained by STEFLUX.**

780